# Chaos, Extremism and Optimism:
# Volume Analysis of Learning in Games

**Yun Kuen Cheung**
Royal Holloway University of London
yunkuen.cheung@rhul.ac.uk

**Georgios Piliouras**
Singapore University of Technology and Design
georgios@sutd.edu.sg

## Abstract

We perform volume analysis of Multiplicative Weights Updates (MWU) and its optimistic variant (OMWU) in zero-sum as well as coordination games. Our analysis provides new insights into these game/dynamical systems, which seem hard to achieve via the classical techniques within Computer Science and ML.

First, we examine these dynamics not in their original space (simplex of actions) but in a dual space (aggregate payoffs of actions). Second, we explore how the volume of a set of initial conditions evolves over time when it is pushed forward according to the algorithm. This is reminiscent of approaches in evolutionary game theory where replicator dynamics, the continuous-time analogue of MWU, is known to preserve volume in all games. Interestingly, when we examine discrete-time dynamics, the choices of the game and the algorithm both play a critical role. So whereas MWU expands volume in zero-sum games and is thus Lyapunov chaotic, we show that OMWU contracts volume, providing an alternative understanding for its known convergent behavior. Yet, we also prove a no-free-lunch type of theorem, in the sense that when examining coordination games the roles are reversed.

Using these tools, we prove two novel, rather negative properties of MWU in zero-sum games. (1) Extremism: even in games with a unique fully-mixed Nash equilibrium, the system recurrently gets stuck near pure-strategy profiles, despite them being clearly unstable from game-theoretic perspective. (2) Unavoidability: given any set of *good* states (with a rather relaxed interpretation of "good" states), the system cannot avoid *bad* states indefinitely.

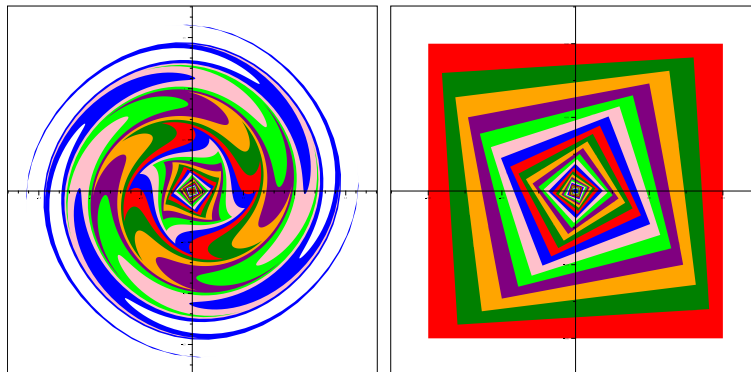

Figure 1: MWU (left) and OMWU (right) in Matching-Pennies game in the dual space. The origin is the unique Nash equilibrium. In these 2-D systems, volume is area. MWU: The initial set is the tiny red square around the equilibrium. When it is evolved via MWU, it rotates tornado-like and its area increases. [11] OMWU: The initial set is the outermost red square. When it is evolved via OMWU, it shrinks toward the equilibrium and its area decreases.

# 1 Introduction

In recent years, fuelled by AI applications such as Generative Adversarial Networks (GANs), there has a been a strong push towards a more detailed understanding of the behavior of online learning dynamics in zero-sum games and beyond. Even when focusing on the canonical case of bilinear zero-sum games, the emergent behavior depends critically on the choice of the training algorithms. Results can macroscopically be grouped in three distinct categories: convergence, divergence and recurrence. Specifically, for most standard regret minimizing dynamics and online optimization dynamics, such as Multiplicative Weights Updates (MWU) [3] or gradient descent [9], although their time averages converge [15], their day-to-day behavior diverges away from Nash equilibria [5, 10]. On the other hand, some game-theoretically inspired dynamics, such as Optimistic Multiplicative Weights Updates (OMWU), converge [13, 12]. (Numerous other convergent heuristics have also been recently analysed, e.g. [22, 18, 19, 7, 1].) Finally, if we simplify learning into continuous-time ordinary differential equations (ODEs), such as replicator dynamics (the continuous time analogue of MWU), the emergent behavior becomes almost periodic (Poincaré recurrence) [24, 23, 8]. This level of complex case-by-case analysis just to understand bilinear zero-sum games seems daunting. Can we find a more principled approach behind these results that is applicable to more general games?

One candidate is *volume analysis*, a commonly used tool in the area of dynamical systems. Effectively what it does is to consider a set of starting points with positive volume (Lebesgue measure), and analyse how the volume changes as the set evolves forward in time. As we shall see, an advantage of volume analysis is its general applicability, for it can be used to analyse not just ODEs but different discrete-time algorithms such as MWU and OMWU in different types of games.

In evolutionary game theory volume analysis has been applied to *continuous-time* dynamical systems (see [20, Section 11], [16, Section 3] and [25, Chapter 9]). Eshel and Akin [14] showed that replicator dynamics in any matrix game is volume preserving in the dual (aggregate payoff) space. This result is in fact a critical step in the proof of Poincaré recurrence in zero-sum games. Intuitively, if we think of the set of initial conditions as our uncertainty about where the starting point is, since uncertainty does not decrease, asymptotic convergence to equilibrium is not possible. Instead, due to physics-like conservation laws [6], the system ends up cycling.

Recently volume analysis has been applied to *discrete-time* learning algorithms in a series of games, including two-person zero-sum games, graphical constant-sum games, generalized Rock-Paper-Scissors games and $2 \times 2$ bimatrix games [11] (see Figure 2 for an illumination of volume expansion of MWU in a graphical constant-sum game). Among other results, MWU in zero-sum games was proven to be *Lyapunov chaotic* in the dual space. The proof relies on establishing that the volume of any set is expanding exponentially fast. Lyapunov chaos is one of the most classical notions in the area of dynamical systems that captures instability and unpredictability. More precisely, it captures the following type of *butterfly effect*: when the starting point of a dynamical system is slightly perturbed, the resulting trajectories and final outcomes diverge quickly. Such systems are very sensitive to round-off errors in computer simulations, raising the need for new discretization schemes, training algorithms [4].

**Our Contributions, and Roadmap for This Paper.** Our contributions can be summarized into two categories, both stemming from volume analysis. First, besides the numerical instability and unpredictability already mentioned, we discover two novel and unfavorable properties of MWU in zero-sum games, which are consequences of exponential volume expansion. We call them *unavoidability* and *extremism*. We have given informal descriptions of these two properties in the abstract; we will give more details about them below.

Second, we carry out volume analysis on OMWU and discover that its volume-changing behavior is in stark contrast with MWU. To understand why we should be interested in such an analysis, we first point out that in the study of game dynamics, a primary goal is to find algorithms that behave well in as many games as possible. Recently, OMWU was shown to achieve stability in zero-sum games, despite its strong similarity with MWU, which is chaotic in zero-sum games. It is natural to ask how whether this stability of OMWU generalizes to other games. We provide a negative answer, by proving that OMWU is volume-expanding and Lyapunov chaotic in coordination games; see Figure 3 for a summary of this no-free-lunch phenomenon. We show that volume is exponentially decreasing for OMWU in zero-sum games, mirroring the recent stability results in the original (primal) space (i.e. simplex of actions) [12, 13].

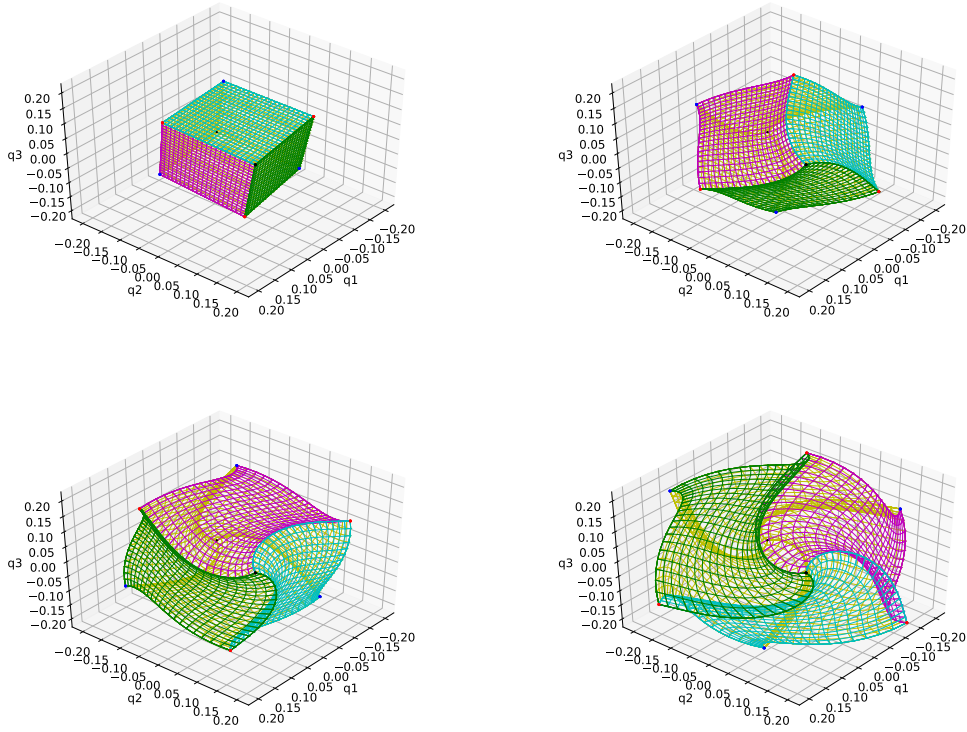

Figure 2: Volume expansion of MWU in a three-player graphical zero-sum game. Players are numbered 0,1,2. Each player has two strategies, Head and Tail. Each edge-game is a Matching Pennies game: Player $i$ wants to match with Player $(i + 1) \mod 3$ but wants to mis-match with Player $(i - 1) \mod 3$. The initial set is a small rectangular box around a Nash equilibrium in the dual space. The figures show snapshots of the initial set (top left), and its evolution after 4500 (top right), 9000 (bottom left) and 13500 (bottom right) steps. An animation is available at http://cs.rhul.ac.uk/~cheung/mwu-graphical-matching-pennies.mp4.

We remark that the volume analysis of OMWU is technically more involved than that of MWU. We define an ODE system that is the continuous-time analogue of OMWU. Our volume analysis relies crucially on the observation that OMWU is an *online Euler discretization* of this ODE system. See Sections 5 and 6 for the details about OMWU.

|      | Zero-sum Games | Coordination Games |
| --- | --- | --- |
| MWU | + [11] | − (see supplementary material E) |
| OMWU | − (see supplementary material E) | + (Theorem 8) |

Figure 3: How volume changes in the dual space. "+" denotes exponential volume expansion, unavoidability and extremism, while "−" denotes exponential volume contraction.

In Section 2, we discuss how volume analysis can be carried out on learning algorithms that are *gradual*, i.e. controlled by a step-size $\epsilon$. We demonstrate that volume analysis can often be reduced to analyzing a polynomial of $\epsilon$ (see Eqn. (6)). This convincingly indicates that volume analysis can be readily applicable to a broad family of learning algorithms. In the rest of this introduction, we discuss extremism and unavoidability in more detail.

**Extremism (Section 4).** Our extremism theorem (Theorem 5) states that given any zero-sum game that satisfies a mild regularity condition, there exists a dense set of starting points from which MWU will lead to a state where both players concentrate their game-plays on only one strategy. More precisely, let $\mathbf{x}, \mathbf{y}$ denote the mixed strategies of the two players. For any $\delta > 0$, there is a dense set of starting points $(\mathbf{x}^0, \mathbf{y}^0)$, from which MWU with a suitably small step-size leads to $(\mathbf{x}^t, \mathbf{y}^t)$ for some time $t$, at which there exists a strategy $j$ with $x_j^t \geq 1 - \delta$, and a strategy $k$ with $y_k^t \geq 1 - \delta$.

To understand how bizarre extremism is, consider the classical Rock-Paper-Scissors (RPS) game, which has a unique fully-mixed Nash equilibrium. The extremism theorem indicates that there exists a starting point arbitrarily close to the equilibrium, which will eventually lead to a situation where each player essentially sticks with one strategy for a long time[1]. As no pure Nash equilibrium exists, the trajectory will recurrently approach and escape such extremal points infinitely often (Theorem 6), demonstrating that the dynamic is rather unstable.

**Unavoidability (Section 3).** The extremism theorem is actually an indirect consequence of an unavoidability theorem of MWU in zero-sum games. Unavoidability is a notion first introduced in a subarea of (automatic) control theory called "avoidance control" [21], which addresses the following type of problems: for dynamical/automatic systems, analyze whether they can always avoid reaching certain *bad* states, e.g. collisions of robots/cars, or places with severe weather conditions.

To explain unavoidability of MWU in general games, we need the notion of *uncontrollability*. Let $U$ be a region in the interior of the primal simplex, and let $V$ be the corresponding set of $U$ in the dual space. Informally, we say $U$ is *uncontrollable* if any subset of $V$ is exponentially volume-expanding in the dual space. As the volume of a set $S \subset V$ expands quickly, it turns out to be impossible for $V$ to fully contain the evolution of $S$ after a long enough time. When converting to the primal space, this implies that primal correspondence of $S$ evolves to escape from $U$ (Theorem 3). When $U$ is thought of as a set of *good* points and its complement contains *bad* points, the punchline is:

*When a good set is uncontrollable, the bad set is unavoidable.*

Note that the above discussion concerns general games. When we narrow down to zero-sum games, the results in [11] indicate that under mild regularity conditions, *any* set $U$ in the strict interior of the primal simplex is uncontrollable. Thus, for MWU in zero-sum games, one can liberally choose an interpretation of "good" states, but the corresponding bad set is unavoidable.

Some ideas behind the proof of unavoidability theorem come from [11], which demonstrated several negative properties of MWU in specific games such as generalized RPS games. Our key contribution is to formulate and prove the fully generalized statement about this property.

## 2  Preliminary

**Games.** In this paper, we focus on two-person general normal-form games. The strategy set of Player $i$ is $S_i$. Let $n = |S_1|$ and $m = |S_2|$. We assume $n, m \geq 2$ throughout this paper. Let $\mathbf{A}$ and $\mathbf{B}$ be two $S_1 \times S_2$ matrices, which represent the payoffs to Players 1 and 2 respectively. We assume all payoffs are bounded within the interval $[-1, 1]$. Let $\Delta^d := \left\{ (z_1, z_2, \cdots, z_d) \in \mathbb{R}^d \mid \sum_{j=1}^d z_j = 1, \text{ and } \forall j, z_j \geq 0 \right\}$. We call $\Delta := \Delta^n \times \Delta^m$ the *primal simplex* or *primal space* of the game, which contains the set of all mixed strategy profiles of the two players. When a zero-sum game is concerned, only the matrix $\mathbf{A}$ needs to be specified, as $\mathbf{B} = -\mathbf{A}$. We say a zero-sum game $(\mathbf{A}, -\mathbf{A})$ is *trivial* if there exist real numbers $a_1, a_2, \cdots, a_n$ and $b_1, b_2, \cdots, b_m$ such that $A_{jk} = a_j + b_k$. A trivial game is not interesting as each player has a clear dominant strategy; for Player 1 it is $\arg\max_{j \in S_1} a_j$, while for Player 2 it is $\arg\min_{k \in S_2} b_k$. Following [11], we measure the distance of a zero-sum game $\mathbf{A}$ from triviality by

$$c(\mathbf{A}) := \min_{a_1, \cdots, a_n, b_1, \cdots, b_m \in \mathbb{R}} \left[ \max_{j \in S_1, k \in S_2} \left( A_{jk} - a_j - b_k \right) - \min_{j \in S_1, k \in S_2} \left( A_{jk} - a_j - b_k \right) \right]. \quad (1)$$

Observe that if $\mathbf{A}'$ is a sub-matrix of $\mathbf{A}$, then $c(\mathbf{A}') \leq c(\mathbf{A})$. If one of the two dimensions of $\mathbf{A}$ is one, then $c(\mathbf{A}) = 0$. By setting all $a_j, b_k$ to zero, we have the trivial bound $c(\mathbf{A}) \leq 2$.

For a coordination game, i.e. a game with payoff matrices in the form of $(\mathbf{A}, \mathbf{A})$, we also measure its distance from triviality using Equation (1).

**MWU and OMWU Update Rules in Dual and Primal Spaces.** As is well-known, MWU and OMWU can be implemented either in the primal space, or in a dual space. The dual space is $\mathcal{D} := \mathbb{R}^n \times \mathbb{R}^m$, in which MWU with positive step size $\epsilon$ generates a sequence of updates $(\mathbf{p}^0, \mathbf{q}^0), (\mathbf{p}^1, \mathbf{q}^1), (\mathbf{p}^2, \mathbf{q}^2), \cdots$, where $p_j^t - p_j^0$ is $\epsilon$ times the cumulative payoff to Player 1's strategy $j$ up to time $t$, and $q_k^t - q_k^0$ is $\epsilon$ times the cumulative payoff to Player 2's strategy $k$ up to time $t$. At each time step, a

point $(\mathbf{p}^t, \mathbf{q}^t) \in \mathcal{D}$ is converted to a point $(\mathbf{x}^t, \mathbf{y}^t) = (\mathbf{x}(\mathbf{p}^t), \mathbf{y}(\mathbf{q}^t)) \in \Delta$ by the following rules:

$$x_j^t = x_j(\mathbf{p}^t) = \exp(p_j^t) \big/ \big(\textstyle\sum_{\ell \in S_1} \exp(p_\ell^t)\big) \ \ \& \ \ y_k^t = y_k(\mathbf{q}^t) = \exp(q_k^t) \big/ \big(\textstyle\sum_{\ell \in S_2} \exp(q_\ell^t)\big) \ . \quad (2)$$

We let $\mathsf{G}$ denote the function that converts a dual point to a primal point, i.e. $\mathsf{G}(\mathbf{p}, \mathbf{q}) = (\mathbf{x}(\mathbf{p}), \mathbf{y}(\mathbf{q}))$.

For MWU in a general-sum game, the payoffs to Player 1's all strategies in round $(t+1)$ is represented by the vector $\mathbf{A} \cdot \mathbf{y}(\mathbf{q}^t)$, while the payoffs to Player 2's all strategies in round $(t+1)$ is represented by the vector $\mathbf{B}^{\mathsf{T}} \cdot \mathbf{x}(\mathbf{p}^t)$. Accordingly, the MWU update rule in the dual space is

$$\mathbf{p}^{t+1} \ = \ \mathbf{p}^t + \epsilon \cdot \mathbf{A} \cdot \mathbf{y}(\mathbf{q}^t) \quad \text{and} \quad \mathbf{q}^{t+1} \ = \ \mathbf{q}^t + \epsilon \cdot \mathbf{B}^{\mathsf{T}} \cdot \mathbf{x}(\mathbf{p}^t). \quad (3)$$

The above update rule in the dual space is equivalent to the following MWU update rule in the primal space with starting point $\mathsf{G}(\mathbf{p}^0, \mathbf{q}^0)$, which some readers might be more familiar with:

$$x_j^{t+1} \ = \ \frac{x_j^t \cdot \exp(\epsilon \cdot [\mathbf{A} \cdot \mathbf{y}^t]_j)}{\sum_{\ell \in S_1} x_\ell^t \cdot \exp(\epsilon \cdot [\mathbf{A} \cdot \mathbf{y}^t]_\ell)} \quad \text{and} \quad y_k^{t+1} \ = \ \frac{y_k^t \cdot \exp(\epsilon \cdot [\mathbf{B}^{\mathsf{T}} \cdot \mathbf{x}^t]_k)}{\sum_{\ell \in S_2} y_\ell^t \cdot \exp(\epsilon \cdot [\mathbf{B}^{\mathsf{T}} \cdot \mathbf{x}^t]_\ell)}. \quad (4)$$

For OMWU in a general-sum game with step-size $\epsilon$, its update rule in the dual space starts with $(\mathbf{p}^0, \mathbf{q}^0) = (\mathbf{p}^1, \mathbf{q}^1)$, and for $t \geq 1$,

$$\mathbf{p}^{t+1} = \mathbf{p}^t + \epsilon \cdot \big[2\mathbf{A} \cdot \mathbf{y}(\mathbf{q}^t) - \mathbf{A} \cdot \mathbf{y}(\mathbf{q}^{t-1})\big] \ \& \ \mathbf{q}^{t+1} = \mathbf{q}^t + \epsilon \cdot \big[2\mathbf{B}^{\mathsf{T}} \cdot \mathbf{x}(\mathbf{p}^t) - \mathbf{B}^{\mathsf{T}} \cdot \mathbf{x}(\mathbf{p}^{t-1})\big], \quad (5)$$

where $\mathbf{x}(\mathbf{p}^t), \mathbf{y}(\mathbf{q}^t)$ are as defined in (2). Note that for the rule (5), for $t \geq 2$, $\mathbf{p}^t - \mathbf{p}^0 = \epsilon(\sum_{\tau=1}^{t-2} \mathbf{A} \cdot \mathbf{y}(\mathbf{q}^\tau) + 2 \cdot \mathbf{A} \cdot \mathbf{y}(\mathbf{q}^{t-1}))$, which is $\epsilon$ times the cumulative payoff to strategy $j$ from time 2 to time $t$, but with a double weight on the last-iterate payoff.

**Relationships between Primal and Dual Spaces.** Here, we clarify some facts about primal and dual spaces and their relationships. Equation (2) provides a conversion from a dual point in $\mathcal{D}$ to a point in the interior of the primal space, i.e. $\mathsf{int}(\Delta)$. It is not hard to see that there exist infinitely many points in $\mathcal{D}$ which convert to the same point in $\mathsf{int}(\Delta)$. By [11, Proposition 1], if $(\mathbf{p}, \mathbf{q}), (\mathbf{p}', \mathbf{q}') \in \mathcal{D}$, then $(\mathbf{x}(\mathbf{p}), \mathbf{y}(\mathbf{q})) = (\mathbf{x}(\mathbf{p}'), \mathbf{y}(\mathbf{q}'))$ if and only if $\mathbf{p} - \mathbf{p}' = c_1 \cdot \mathbf{1}$ and $\mathbf{q} - \mathbf{q}' = c_2 \cdot \mathbf{1}$ for some $c_1, c_2 \in \mathbb{R}$. For any $S \subset \mathsf{int}(\Delta)$, let $\mathsf{G}^{-1}(S)$ denote the set of points $(\mathbf{p}, \mathbf{q}) \in \mathcal{D}$ such that $\mathsf{G}(\mathbf{p}, \mathbf{q}) \in S$.

Since the primal and dual spaces are not in one-one correspondence, some readers might argue that the *reduced* dual space used by Eshel and Akin [14] (in which its $(n + m - 2)$ dual variables denote the quantities $p_1 - p_n, p_2 - p_n, \cdots, p_{n-1} - p_n, q_1 - q_m, q_2 - q_m, \cdots, q_{m-1} - q_m$) is a better choice. Our reason for choosing $\mathcal{D}$ as the dual space to work with is simply because we are unable to establish the same type of results (like Lemma 2 below) for the reduced dual space.

While we use dual volume as the mean for analysis, when measuring instability, what we really care is the diameter of the dual set or its corresponding primal set. (The diameter of a set is the maximum $\ell_2$-distance between any two points in the set.) The following proposition shows that volume expansion in the dual space implies large diameter in the primal space, if there is a primal point bounded away from the simplex boundary.

**Proposition 1.** *Let $S$ be a set in the dual space with Lebesgue measure (i.e. volume) $v$. Suppose there exists $j \in S_1, k \in S_2$ such that $\max_{(\mathbf{p}, \mathbf{q}) \in S} p_j - \min_{(\mathbf{p}, \mathbf{q}) \in S} p_j \leq R_j$ and $\max_{(\mathbf{p}, \mathbf{q}) \in S} q_k - \min_{(\mathbf{p}, \mathbf{q}) \in S} q_k \leq R_k$. Also, suppose that for some $\kappa > 0$, there exists a point $(\mathbf{x}, \mathbf{y}) \in \mathsf{G}(S)$ such that either every entry of $\mathbf{x}$ is at least $\kappa$ or every entry of $\mathbf{y}$ is at least $\kappa$. Then the diameter of $\mathsf{G}(S)$ is at least $\big[1 - \exp\big(-(1/4) \cdot (v/R_j R_k)^{1/(n+m-2)}\big)\big] \cdot \kappa$.*

In the supplementary material A, we present concrete examples to show that (A) volume contraction in the dual space does *not* necessarily imply stability in either the dual or the primal space; and (B) volume expansion in the dual space does *not* necessarily imply instability in the primal space if the primal set is near the simplex boundary.

**Dynamical System, Jacobian, and Volume of Flow.** We consider discrete-time dynamical systems in $\mathbb{R}^d$. Such a dynamical system is determined recursively by a starting point $\mathbf{s}(0) \in \mathbb{R}^d$ and an update rule of the form $\mathbf{s}(t+1) = G(\mathbf{s}(t))$, for some function $G : \mathbb{R}^d \to \mathbb{R}^d$. Here, we focus on the special case when the update rule is *gradual*, i.e. it is in the form of $\mathbf{s}(t+1) = \mathbf{s}(t) + \epsilon \cdot F(\mathbf{s}(t))$, where $F : \mathbb{R}^d \to \mathbb{R}^d$ is a smooth function and step-size $\epsilon > 0$. When $F$ and $\epsilon$ are given, the flow of the starting point $\mathbf{s}(0)$ at time $t$, denoted by $\Phi(t, \mathbf{s}(0))$, is simply the point $\mathbf{s}(t)$ generated by the above recursive update rule; the flow of a set $S \subset \mathbb{R}^d$ at time $t$ is $\Phi(t, S) := \{\Phi(t, \mathbf{s}) \mid \mathbf{s} \in S\}$. Since $F$ does not depend on time $t$, $\Phi(t_1 + t_2, S) = \Phi(t_2, \Phi(t_1, S))$ for all $t_1, t_2 \in \mathbb{N}$.

By equipping $\mathbb{R}^d$ with the standard Lebesgue measure, the *volume* of a measurable set $S$, denoted by $\text{vol}(S)$, is simply its measure. Given a bounded and measurable $S \subset \mathbb{R}^d$, if the discrete flow in one time step maps $S$ to $S' = \Phi(1, S)$ injectively, then by integration by substitution for multi-variables,

$$\text{vol}(S') \;=\; \int_{\mathbf{s} \in S} \det\left(\mathbf{I} + \epsilon \cdot \mathbf{J}(\mathbf{s})\right)\,dV, \;\; \text{where} \;\; \mathbf{J}(\mathbf{s}) = \begin{bmatrix} \frac{\partial}{\partial s_1} F_1(\mathbf{s}) & \frac{\partial}{\partial s_2} F_1(\mathbf{s}) & \cdots & \frac{\partial}{\partial s_d} F_1(\mathbf{s}) \\ \vdots & \vdots & \ddots & \vdots \\ \frac{\partial}{\partial s_1} F_d(\mathbf{s}) & \frac{\partial}{\partial s_2} F_d(\mathbf{s}) & \cdots & \frac{\partial}{\partial s_d} F_d(\mathbf{s}) \end{bmatrix}, \quad (6)$$

and $\mathbf{I}$ is the identity matrix. $\mathbf{J}(\mathbf{s})$ is called the *Jacobian* matrix.

Clearly, analyzing the determinant in the integrand in (6) is crucial in volume analysis; we call it the *volume integrand*. When the determinant is expanded using the Leibniz formula, it becomes a polynomial of $\epsilon$, in the form of $1 + C(\mathbf{s}) \cdot \epsilon^h + \mathcal{O}(\epsilon^{h+1})$ for some integer $h \geq 1$. Thus, when the step-size $\epsilon$ is sufficiently small, the sign of $C(\mathbf{s})$ dictates on whether the volume expands or contracts.

In our case, $\mathbf{s}$ refers to a cumulative payoff vector $(\mathbf{p}, \mathbf{q})$. For the MWU update rule (3) in the dual space, the volume integrand can be written as $1 + C_{(\mathbf{A},\mathbf{B})}(\mathbf{p}, \mathbf{q}) \cdot \epsilon^2 + \mathcal{O}(\epsilon^4)$ [11], where

$$C_{(\mathbf{A},\mathbf{B})}(\mathbf{p}, \mathbf{q}) \;=\; -\sum_{j \in S_1} \sum_{k \in S_2} x_j(\mathbf{p}) \cdot y_k(\mathbf{q}) \cdot (A_{jk} - [\mathbf{A} \cdot \mathbf{y}(\mathbf{q})]_j) \cdot (B_{jk} - [\mathbf{B}^\mathsf{T} \cdot \mathbf{x}(\mathbf{p})]_k). \quad (7)$$

Note that $C_{(\mathbf{A},\mathbf{B})}(\mathbf{p}, \mathbf{q})$ depends on the primal variables $\mathbf{x}(\mathbf{p}), \mathbf{y}(\mathbf{q})$ but not explicitly on $\mathbf{p}, \mathbf{q}$. Thus, it is legitimate to refer to this value using the primal variables as input parameters to $C_{(\mathbf{A},\mathbf{B})}$, i.e., we can refer to its value by $C_{(\mathbf{A},\mathbf{B})}(\mathbf{x}, \mathbf{y})$ too. [11] showed the following lemma.

**Lemma 2.** *[11, Lemma 3, Section 4.1 and Appendix B] The following hold: (1) In any two-person zero-sum game $(\mathbf{A}, -\mathbf{A})$, at any point $(\mathbf{x}, \mathbf{y}) \in \Delta$, $C_{(\mathbf{A},-\mathbf{A})}(\mathbf{x}, \mathbf{y}) \geq 0$. Indeed, $C_{(\mathbf{A},-\mathbf{A})}(\mathbf{x}, \mathbf{y})$ equals to the variance of the random variable $X$ such that $X = (A_{jk} - [\mathbf{A}\mathbf{y}]_j - [\mathbf{A}^\mathsf{T}\mathbf{x}]_k)$ with probability $x_j y_k$, for all $(j, k) \in S_1 \times S_2$. (2) When $\epsilon \leq 1/4$, the update rule (3) in the dual space is injective. (3) When $\epsilon < \min\left\{1/(32n^2m^2), C_{(\mathbf{A},\mathbf{B})}(\mathbf{p}, \mathbf{q})\right\}$, the volume integrand at point $(\mathbf{p}, \mathbf{q})$ is lower bounded by $1 + (C_{(\mathbf{A},\mathbf{B})}(\mathbf{p}, \mathbf{q}) - \epsilon)\epsilon^2$. Thus, in (6), if $\overline{C} := \inf_{(\mathbf{p},\mathbf{q}) \in S} C_{\mathbf{A},\mathbf{B}}(\mathbf{p}, \mathbf{q}) > 0$, then for all $0 < \epsilon \leq \min\left\{1/(32n^2m^2), \overline{C}\right\}$, $\text{vol}(S') \geq \left[1 + \left(\overline{C} - \epsilon\right)\epsilon^2\right] \cdot \text{vol}(S)$.*

By the definition of $C_{(\mathbf{A},\mathbf{B})}$, it is straightforward to see that

$$C_{(\mathbf{A},\mathbf{A})}(\mathbf{p}, \mathbf{q}) \;=\; -C_{(\mathbf{A},-\mathbf{A})}(\mathbf{p}, \mathbf{q}). \quad (8)$$

Thus, for any coordination game $(\mathbf{A}, \mathbf{A})$, and for any $(\mathbf{p}, \mathbf{q}) \in \mathcal{D}$, $C_{(\mathbf{A},\mathbf{A})}(\mathbf{p}, \mathbf{q}) \leq 0$ due to Lemma 2.

**Lyapunov Chaos.** In the study of dynamical systems, *Lyapunov chaos* generally refers to the phenomenon where a tiny difference in the starting points can yield widely diverging outcomes *quickly*. A classical measure of chaos is the *Lyapunov time*, defined as: when the starting point is perturbed by a distance of tiny $\gamma$, for how long will the trajectories of the two starting points remain within a distance of at most $2\gamma$. [11] showed that if the volume of a set increases at a rate of $\Omega((1 + \beta)^t)$, its diameter increases at a rate of at least $\Omega((1 + \beta/d)^t)$, where $d$ is the dimension of the system, thus indicating that the Lyapunov time is at most $\mathcal{O}(d/\beta)$.

## 3 Unavoidability of MWU in Games

The result in this section holds for MWU in general games. Recall the definition of $C_{(\mathbf{A},\mathbf{B})}(\mathbf{p}, \mathbf{q})$ in Equation (7), and the discussion on extending its definition to the primal space (i.e. $C_{(\mathbf{A},\mathbf{B})}(\mathbf{x}, \mathbf{y})$) below Equation (7). To avoid clutter, when the underlying game $(\mathbf{A}, \mathbf{B})$ is clear from context, we write $C(\cdot)$ for $C_{(\mathbf{A},\mathbf{B})}(\cdot)$. Theorem 3 is the *unavoidability theorem*, the main theorem in this section.

**Definition 1.** *A set $U$ is called a primal open set if it is an open set, and it is a subset of $\text{int}(\Delta)$. A primal open set $U$ is* uncontrollable *if $\inf_{(\mathbf{x},\mathbf{y}) \in U} C(\mathbf{x}, \mathbf{y}) > 0$. Given a primal open set $U$, we say $U'$ is a dense subset of $U$ if for any open ball $B \subset U$, $B \cap U'$ is non-empty.*

**Theorem 3.** *Let $U$ be an uncontrollable primal open set with $\overline{C} := \inf_{(\mathbf{x},\mathbf{y}) \in U} C(\mathbf{x}, \mathbf{y}) > 0$. If the step-size $\epsilon$ in the update rule (4) satisfies $\epsilon < \min\left\{\frac{1}{32n^2m^2}, \overline{C}\right\}$, then there exists a dense subset of $U$ such that the flow of each such point must eventually reach a point outside $U$.*

One perspective to think about the unavoidability theorem is to consider $U$ as a collection of good points, while $\Delta \setminus U$ is the set of bad points that we want to *avoid*. We desire the game dynamic to

stay within $U$ forever, when the starting point is in $U$. The theorem then presents a negative property, which states that if $U$ is uncontrollable, then there is a dense subset of $U$ such that if we start from any point in the dense subset, the game dynamic must eventually reach a point that we want to avoid.

In particular, when the underlying game is a zero-sum game, due to Lemma 2, $\inf_{(\mathbf{x},\mathbf{y})\in U} C(\mathbf{x},\mathbf{y}) \geq 0$ for *any* $U$. With some mild assumptions on $U$ and the underlying game, it is foreseeable that the infimum becomes strictly positive, for which Theorem 3 is applicable. For instance, if the zero-sum game $(\mathbf{A}, -\mathbf{A})$ is not trivial (i.e. the measure $c(\mathbf{A})$ in (1) is positive) and $U$ collects all points $(\mathbf{x}, \mathbf{y})$ such that all $x_j, y_k > \delta$ for some fixed $\delta > 0$, then the infimum is strictly positive due to Lemma 2 Part (1); see [11] for a detailed explanation. Thus, for quite general scenarios, MWU in zero-sum game *cannot* avoid bad states, regardless of what "good" or "bad" really mean.

Next, we discuss the proof of Theorem 3. In Definition 1, we have defined uncontrollability of a set in the primal space. In the dual space, the definition of uncontrollability is similar: an open set $V \subset \mathcal{D}$ is uncontrollable if $\inf_{(\mathbf{p},\mathbf{q})\in V} C(\mathbf{p},\mathbf{q}) > 0$. Lemma 4 below is the key to proving Theorem 3. Its statement is a bit technical, so we give an informal description of it. Suppose $V$ is uncontrollable in $\mathcal{D}$, and $S$ is a *bounded* subset of $V$ with positive volume. Then the volume of the flow $\Phi(t, S)$ increases exponentially with $t$ so long as $\Phi(t, S)$ remains in $V$. Then we use the observation that this quick exponential volume growth cannot occur indefinitely to show that eventually $\Phi(t, S)$ must *escape* from $V$, i.e. $\Phi(\tau, S) \not\subset V$ for some large enough $\tau$. Theorem 3 follows quite readily by considering $V$ as the dual correspondence set of $U$, i.e. $V = \mathsf{G}^{-1}(U)$.

**Lemma 4.** *Let $V$ be an uncontrollable open set in the dual space, with $\overline{C} := \inf_{(\mathbf{p},\mathbf{q})\in V} C(\mathbf{p},\mathbf{q}) > 0$. Assume that the step-size $\epsilon$ in the update rule (3) satisfies $0 < \epsilon < \min\left\{\frac{1}{32n^2m^2}, \overline{C}\right\}$. Let $S \subset V$ be a measurable set with positive volume, and let $d(S)$ denote the $\ell_\infty$-diameter of $S$. Then there exists $\tau$ with $\tau \leq \max\left\{\frac{d(S)}{2\epsilon}, \frac{8(n+m)}{(\overline{C}-\epsilon)\epsilon^2}\ln\frac{4(n+m)}{(\overline{C}-\epsilon)\epsilon^2}, \frac{4}{(\overline{C}-\epsilon)\epsilon^2}\ln\frac{1}{\mathsf{vol}(S)}\right\}$, such that the flow of $S$ at time $\tau$, i.e. $\Phi(\tau, S)$, contains a point which is not in $V$.*

## 4 Extremism of MWU in Zero-Sum Games

Here, we focus on MWU in zero-sum games. [5, 10] showed that the dynamic converges to the boundary of $\Delta$ and fluctuates bizarrely near the boundary, by using a potential function argument. However, the potential function has value $+\infty$ at every point on the boundary, so it cannot provide useful insight on how the dynamic behaves near the boundary. In general, the behavior near boundary can be highly unpredictable, as suggested by the "chaotic switching" phenomena found in [2], although more regular (yet still surprising) patterns were found in lower-dimensional systems [17].

A central discouraging message in [5, 10] is that convergence towards the boundary of $\Delta$ is inevitable even when the underlying zero-sum game has a fully-mixed Nash equilibrium. What can we still hope for after this? Will $(\mathbf{x}^t, \mathbf{y}^t)$ remain within a somewhat *reasonable* range forever? We give a strikingly negative answer to this question for almost all zero-sum games with the theorems below.

**Definition 2.** *The* extremal domain with threshold $\delta$ *consists of all points $(\mathbf{x}, \mathbf{y})$ such that each of $\mathbf{x}, \mathbf{y}$ has exactly one entry of value at least $1 - \delta$.*

**Theorem 5.** *Let $(\mathbf{A}, -\mathbf{A})$ be a two-person zero-sum game. Suppose the following two conditions hold: (A) Every $2 \times 2$ sub-matrix of $\mathbf{A}$ is non-trivial. Let $\alpha_1 > 0$ denote the minimum distance from triviality of all $2 \times 2$ sub-matrices of $\mathbf{A}$ (recall the distance measure (1).); and (B) No two entries in the same row or the same column have exactly the same value. Let $\alpha_2 > 0$ be the minimum difference between any two entries of $\mathbf{A}$ in the same row or same column.*

*Let $N := \max\{n, m\}$. For any $\delta < \alpha_2/4$, if both players use MWU with step-size $\epsilon$ satisfying $\epsilon < \min\left\{\frac{1}{32n^2m^2}, \frac{(\alpha_1)^2}{18}\cdot\left(\frac{\delta}{N-1}\right)^{8(N-1)/(\alpha_2-4\delta)+2}\right\}$, then there is a dense subset of $\mathsf{int}(\Delta)$, such that the flow of each such point eventually reaches the extremal domain with threshold $\delta$.*

**Theorem 6.** *Let $v := \max_{\mathbf{x}\in\Delta^n}\min_{\mathbf{y}\in\Delta^m}\mathbf{x}^\mathsf{T}\mathbf{A}\mathbf{y}$ denote the game value of the zero-sum game $(\mathbf{A}, -\mathbf{A})$. In addition to the conditions required in Theorem 5, if (i) $\min_{j\in S_1, k\in S_2}|A_{jk} - v| \geq r > 0$, and (ii) $6\epsilon + 4\delta \leq r$, then there exists a dense subset of $\mathsf{int}(\Delta)$, such that the flow of each such point visits and leaves extremal domain with threshold $\delta$ infinitely often.*

To see the power of the Theorem 5, consider a zero-sum game with a fully-mixed Nash Equilibrium. The theorem implies that in *any arbitrarily small neighbourhood* of the equilibrium, there is a

starting point such that its flow eventually reaches a point where each player concentrates her game-play on only one strategy. We call this *extremism of game-play*, since both players are single-minded at this point: they concentrate on one strategy and essentially ignoring all other strategies.

There are two assumptions on the matrix $\mathbf{A}$. If the matrix is drawn uniformly randomly from the space $[-1, +1]^{n \times m}$, it satisfies assumptions (A) and (B) almost surely. Unfortunately, the classical Rock-Paper-Scissors game does not satisfy assumption (A). In the supplementary material C.1, we provide a separate proof which shows a similar result to Theorem 5 for this specific game.

## 5    Continuous Analogue of OMWU

As the OMWU update rule (5) at time $t + 1$ depends on the past updates at times $t$ and $t - 1$, we cannot apply (6) directly for its volume analysis. There are some alternative approaches for this situation, but as we explain in supplementary material D.1 that they do not provide a satisfactory solution. To bypass the issue, we first derive a continuous analogue of OMWU in games as an ODE system. We will use it to derive a clean volume analysis for OMWU in the next section.

**Continuous Analogue of OMWU in General Contexts.**  We focus on Player 1 who uses the OMWU (5). To set up for general contexts, we replace $\mathbf{A} \cdot \mathbf{y}(\mathbf{q}^t)$ by $\mathbf{u}(t)$, which represents the utility (or payoff) vector at time $t$. We assume $\mathbf{u}(t)$ is $C^2$-differentiable. We rewrite the rule as below:

$$\frac{\mathbf{p}^{t+1} - \mathbf{p}^t}{\epsilon} = \mathbf{u}(t) + \epsilon \cdot \frac{(\mathbf{u}(t) - \mathbf{u}(t-1))}{\epsilon}. \tag{9}$$

Recall that for any smooth function $f : \mathbb{R} \to \mathbb{R}$, its first derivative is $\lim_{\epsilon \to 0}(f(x + \epsilon) - f(x))/\epsilon$. For readers familiar with Euler discretization and finite-difference methods, the above discrete-time rule naturally motivates the following differential equation, where for any variable $v$, $\dot{v} \equiv \frac{dv}{dt}$:

$$\dot{\mathbf{p}} = \mathbf{u} + \epsilon \cdot \dot{\mathbf{u}}. \tag{10}$$

To numerically simulate (10), note that in some contexts, only oracle access to $\mathbf{u}(t)$ is available, but $\dot{\mathbf{u}}(t)$ is not directly accessible. For instance, in the context of online learning, at time $N \cdot \Delta t$, the players have only observed $\mathbf{u}(0), \mathbf{u}(\Delta t), \cdots, \mathbf{u}(N \cdot \Delta t)$, but they do not have any knowledge on the *future* values of $\mathbf{u}$. Due to this constraint on information, we have to settle with the backward finite-difference method to approximate $\dot{\mathbf{u}}(N \cdot \Delta t)$:

$$\dot{\mathbf{u}}(N \cdot \Delta t) = \frac{\mathbf{u}(N \cdot \Delta t) - \mathbf{u}((N-1) \cdot \Delta t)}{\Delta t} + \mathcal{O}(\Delta t).$$

Euler method with step-size $\Delta t = \epsilon$ which makes use of the above approximation gives the rule (9), by identifying $\mathbf{p}(t + \epsilon)$ as $\mathbf{p}^{t+1}$. Due to an error that occurs when we approximate $\dot{\mathbf{u}}$ as above,

$$\epsilon \cdot \mathbf{u}(t) + \epsilon \cdot (\mathbf{u}(t) - \mathbf{u}(t-1)) = \epsilon\,[\mathbf{u}(t) + \epsilon \cdot \dot{\mathbf{u}}(t)] + \mathcal{O}(\epsilon^3), \tag{11}$$

where the LHS is the quantity $\mathbf{p}^{t+1} - \mathbf{p}^t$ in the update rule (9), and the first term in the RHS is the standard Euler discretization of (10).

**Proposition 7.** *For ODE system* (10)*, when only* online value oracle *for a $C^2$-differentiable function $\mathbf{u}$ is given, the OMWU rule* (5) *is obtained by first using backward finite-difference with step-size $\epsilon$ to approximate $\dot{\mathbf{u}}$, then applying the Euler discretization with step-size $\epsilon$. Also, Equation* (11) *holds.*

In supplementary material D.2, we discuss some other contexts of computing/approximating $\dot{\mathbf{u}}(t)$.

**Continuous Analogue of OMWU in Games.**  Next, we use (10) to derive an ODE analogue for OMWU in general-sum games. In these and also many other learning contexts, $\mathbf{u}, \dot{\mathbf{u}}$ depend on the driving variables $\mathbf{p}, \mathbf{q}$. In (10), for Player 1, we replace $\mathbf{u}(t)$ by $\mathbf{A} \cdot \mathbf{y}(\mathbf{q}^t)$. By the chain rule,

$$\dot{p}_j = [\mathbf{A} \cdot \mathbf{y}(\mathbf{q})]_j + \epsilon \cdot \frac{d[\mathbf{A} \cdot \mathbf{y}(\mathbf{q})]_j}{dt} = [\mathbf{A} \cdot \mathbf{y}(\mathbf{q})]_j + \epsilon \cdot \sum_{k \in S_2} \frac{\partial[\mathbf{A} \cdot \mathbf{y}(\mathbf{q})]_j}{\partial q_k} \cdot \dot{q}_k.$$

Recall from [11, Equation (7)] that $\frac{\partial[\mathbf{A} \cdot \mathbf{y}(\mathbf{q})]_j}{\partial q_k} = y_k(\mathbf{q}) \cdot (A_{jk} - [\mathbf{A} \cdot \mathbf{y}(\mathbf{q})]_j)$. Thus,

$$\dot{p}_j = [\mathbf{A} \cdot \mathbf{y}(\mathbf{q})]_j + \epsilon \sum_{k \in S_2} y_k(\mathbf{q}) \cdot (A_{jk} - [\mathbf{A} \cdot \mathbf{y}(\mathbf{q})]_j) \cdot \dot{q}_k. \tag{12}$$

Analogously,     $$\dot{q}_k = [\mathbf{B}^\mathsf{T} \cdot \mathbf{x}(\mathbf{p})]_k + \epsilon \sum_{j \in S_1} x_j(\mathbf{p}) \cdot (B_{jk} - [\mathbf{B}^\mathsf{T} \cdot \mathbf{x}(\mathbf{p})]_k) \cdot \dot{p}_j. \tag{13}$$

Formally, the above two formulae, which are in a recurrence format, have *not* yet formed an ODE system. To settle this issue, in supplementary material D.3, we show that when $\epsilon$ is small enough, they can be reduced to a standard ODE system. This formally permits us to use (12) and (13) in the analysis below, as is standard in formal power series when dealing with generating functions.

## 6 Volume Analysis of OMWU in Games

Iterating the recurrence (12) and (13) yields

$$\dot{p}_j = [\mathbf{A} \cdot \mathbf{y}(\mathbf{q})]_j + \epsilon \sum_{k \in S_2} y_k(\mathbf{q}) \cdot (A_{jk} - [\mathbf{A} \cdot \mathbf{y}(\mathbf{q})]_j) \cdot [\mathbf{B}^\mathsf{T} \cdot \mathbf{x}(\mathbf{p})]_k + \mathcal{O}(\epsilon^2);$$
$$\dot{q}_k = [\mathbf{B}^\mathsf{T} \cdot \mathbf{x}(\mathbf{p})]_k + \epsilon \sum_{j \in S_1} x_j(\mathbf{p}) \cdot (B_{jk} - [\mathbf{B}^\mathsf{T} \cdot \mathbf{x}(\mathbf{p})]_k) \cdot [\mathbf{A} \cdot \mathbf{y}(\mathbf{q})]_j + \mathcal{O}(\epsilon^2). \quad (14)$$

Proposition 7 establishes that in general contexts, (5) is the online Euler discretization of (10). As a special case in games, (5) is the online Euler discretization of the recurrence system (12) and (13), and hence of the ODE system (14). Via Equations (14) and (11), we can rewrite (5) as

$$p_j^{t+1} = p_j^t + \epsilon [\mathbf{A}\mathbf{y}(\mathbf{q}^t)]_j + \epsilon^2 \sum_{k \in S_2} y_k(\mathbf{q}^t) \cdot (A_{jk} - [\mathbf{A}\mathbf{y}(\mathbf{q}^t)]_j) \cdot [\mathbf{B}^\mathsf{T}\mathbf{x}(\mathbf{p}^t)]_k + \mathcal{O}(\epsilon^3);$$
$$q_k^{t+1} = q_k^t + \epsilon [\mathbf{B}^\mathsf{T}\mathbf{x}(\mathbf{p}^t)]_k + \epsilon^2 \sum_{j \in S_1} x_j(\mathbf{p}^t) \cdot (B_{jk} - [\mathbf{B}^\mathsf{T}\mathbf{x}(\mathbf{p}^t)]_k) \cdot [\mathbf{A}\mathbf{y}(\mathbf{q}^t)]_j + \mathcal{O}(\epsilon^3). \quad (15)$$

Update rule (5) can be implemented by the players in a distributed manner, but it is hard to be used for volume analysis. In contrast, update rule (15) *cannot* be implemented by the players in distributed manner, since Player 1 does not know the values of $y_k$ and $[\mathbf{B}^\mathsf{T}\mathbf{x}(\mathbf{p})]_k$. However, it permits us to perform a clean volume analysis, since its RHS involves only $\mathbf{p}^t, \mathbf{q}^t$ but not $\mathbf{p}^{t-1}, \mathbf{q}^{t-1}$. In supplementary material E, we compute the volume integrand (recall (6)) of system (15), and show that it is

$$1 - C_{(\mathbf{A},\mathbf{B})}(\mathbf{p},\mathbf{q}) \cdot \epsilon^2 + \mathcal{O}(\epsilon^3). \quad (16)$$

**Lyapunov Chaos of OMWU in Coordination Games.** At this point, it is important to address the similarity of MWU in zero-sum game $(\mathbf{A}, -\mathbf{A})$ and OMWU in coordination game $(\mathbf{A}, \mathbf{A})$. Recall from [11] that the volume integrand for MWU in the zero-sum game is $1 + C_{(\mathbf{A},-\mathbf{A})}(\mathbf{p},\mathbf{q}) \cdot \epsilon^2 + \mathcal{O}(\epsilon^4)$, while by (16) and (8), the volume integrand for OMWU in the coordination game is $1 - C_{(\mathbf{A},\mathbf{A})}(\mathbf{p},\mathbf{q}) \cdot \epsilon^2 + \mathcal{O}(\epsilon^3) = 1 + C_{(\mathbf{A},-\mathbf{A})}(\mathbf{p},\mathbf{q}) \cdot \epsilon^2 + \mathcal{O}(\epsilon^3)$. When $\epsilon$ is sufficiently small, their volume-changing behavior are almost identical. Thus, we can deduce all the Lyapunov chaos, unavoidability and extremism results in Section 3 for OMWU in coordination games. We also have volume contraction results for OMWU in zero-sum game and MWU in coordination game in the full paper. Let $\mathcal{E}_{2,2}^\delta$ be the collection of all points $(\mathbf{x}, \mathbf{y})$, such that at least two entries in $\mathbf{x}$ are larger than $\delta$, and at least two entries in $\mathbf{y}$ are larger than $\delta$.

**Theorem 8.** *Suppose the underlying game is a non-trivial coordination game $(\mathbf{A}, \mathbf{A})$, and the parameter $\alpha_1$ as defined in Theorem 5 is strictly positive. For any $1/2 > \delta > 0$, for any sufficiently small $0 < \epsilon \leq \bar{\epsilon}$ where the upper bound depends on $\delta$, and for any set $S = S(0) \subset \mathsf{G}^{-1}(\mathcal{E}_{2,2}^\delta)$ in the dual space, if $S$ is evolved by the OMWU update rule (5) and if its flow remains a subset of $\mathsf{G}^{-1}(\mathcal{E}_{2,2}^\delta)$ for all $t \leq T-1$, then $\mathrm{vol}(\Phi(T,S)) \geq \left(1 + \epsilon^2 \delta^2 (\alpha_1)^2/4\right)^T \cdot \mathrm{vol}(S)$. Consequently, the system is Lyapunov chaotic within $\mathsf{G}^{-1}(\mathcal{E}_{2,2}^\delta)$ of the dual space, with Lyapunov time $\mathcal{O}((n+m)/(\epsilon^2 \delta^2 (\alpha_1)^2))$.*

**Negative Consequences of Volume Expansion of OMWU in Coordination Game.** In Sections 3 and 4, the unavoidability and extremism theorems are proved via volume expansion. For the extremism theorem, it requires several additional arguments that seem specific to MWU, but these additional arguments work for OMWU too (with very minor modifications). Thus, the unavoidability and extremism theorems hold for OMWU too, after suitably modifying the condition needed for volume expansion and upper bounds on step-sizes.

Suppose a coordination game has a non-pure Nash equilibrium (i.e. a Nash equilibrium $(\mathbf{x}^*, \mathbf{y}^*)$ in which the supports of $\mathbf{x}^*, \mathbf{y}^*$ are both of size at least 2). By the OMWU analogue of Theorem 5, for any tiny open ball $B$ around the equilibrium, there is a dense subset of $\mathrm{int}(\Delta) \cap B$ such that the flow of this point eventually reaches close to an extremal point. In other words, there are points arbitrarily close to the equilibrium with their flows reaching extremal points, i.e. the flows not only move away from the equilibrium *locally*, but they move away for a big distance. This kind of *global instability* result can be applied quite broadly, as many coordination games have non-pure Nash equilibria. For instance, consider a two-player coordination game where each player has $n$ strategies. When both players choose strategy $i$, they both earn \$$A_i$; otherwise they both lose \$$Z$, where $A_i > 0$ and $Z \geq 0$. Then the game has a non-pure Nash equilibrium $(\mathbf{x}^*, \mathbf{x}^*)$, where $x_i^* = \frac{1}{A_i+Z} \Big/ \left(\sum_j \frac{1}{A_j+Z}\right)$, which is strictly positive for all $i$.

## Broader Impact

We do not see any ethical or future societal consequences of this work.

## Acknowledgments and Disclosure of Funding

G. Piliouras gratefully acknowledges AcRF Tier-2 grant (Ministry of Education - Singapore) 2016-T2-1-170, grant PIE-SGP-AI-2018-01, NRF2019-NRF-ANR095 ALIAS grant and NRF 2018 Fellowship NRF-NRFF2018-07 (National Research Foundation Singapore). Y. K. Cheung acknowledges Singapore NRF 2018 Fellowship NRF-NRFF2018-07.

## Footnotes

[1] When $x_j^t < \delta$, it takes $\Omega\left( \frac{1}{\epsilon} \ln \frac{1}{\delta} \right)$ time before $x_j$ can possibly resume a "normal" value, say above $1/20$.

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
