[Supplementary Material]

# Supplementary Materials for the Paper "Chaos, Extremism and Optimism: Volume Analysis of Learning in Games"

Figure 4: Evolution of MWU (left) and OMWU (right) in zero-sum (top) and coordination (bottom) games in the dual space of Eshel and Akin. The initial set is the red square. The top two figures were already shown and discussed in the first page. The bottom two figures correspond to MWU and OMWU in the coordination game $(\mathbf{A}, \mathbf{A})$, where $\mathbf{A}$ is the $2 \times 2$ identity matrix. The vector fields associated with MWU and OMWU are very similar, and so does the two figures. However, when we compute how the areas change, we observe that for MWU, the area is shrinking slowly (from red to blue), while for OMWU, the area is increasing slowly.

Figure 5: Let $\mathbf{x}, \mathbf{y}$ denote respectively the mixed strategies of Players 1 and 2 in the classical Rock-Paper-Scissors game. We plot the quantity $\sum_{j=1}^{3}(x_j)^4 + \sum_{k=1}^{3}(y_k)^4$ against time steps between $1.97 \times 10^6$ to $2.00 \times 10^6$, when both players employ MWU with step-size $\epsilon = 0.005$, and starting point $\mathbf{x}^0 \propto (1, 1, \exp(1/2))$ and $\mathbf{y}^0 \propto (1, 1, \exp(-1/2))$. When the red curve is above the blue horizontal line, *extremism* occurs, i.e., each player concentrate on one strategy, with some $x_j, y_k \geq 0.995$. Within the 30000 time steps, extremism occurs for 22 periods; each period has length around 350.

## A    Missing Examples and Proof in the Preliminary Section

We point out two facts.

(A) volume contraction in the dual space does *not* necessarily imply stability in either the dual or the primal space;

(B) volume expansion in the dual space does *not* necessarily imply instability in the primal space.

To see why (A) is true, consider the following parameterized rectangular set $S(z)$ around the origin in the dual space:

$$S(z) := \{(\mathbf{p}, \mathbf{q}) \in \mathbb{R}^2 \times \mathbb{R}^2 \mid |p_1|, |q_1| \leq 1/z, \ |p_2|, |q_2| \leq \sqrt{z}\}.$$

As $z$ increases, the volume of $S(z) = 1/z$ decreases, but its diameter and the quantities $\max\{p_2 - p_1\}$, $\max\{q_2 - q_1\}$ are $\Theta(\sqrt{z})$ which increase with $z$. Also, since $S$ contains the points

$$((0, \sqrt{z}), (0, \sqrt{z})), ((0, -\sqrt{z}), (0, -\sqrt{z})),$$

when the set $S(z)$ is converted to the primal space, $\mathsf{G}(S)$ contains points close to

$$((0, 1), (0, 1)), ((1, 0), (1, 0))$$

as $z \to \infty$, so the diameter of $\mathsf{G}(S)$ increases to 2 as $z \to \infty$.

To see why (B) is true, consider the following parameterized set $S(z)$ in the dual space:

$$S(z) := \{(\mathbf{p}, \mathbf{q}) \in \mathbb{R}^2 \times \mathbb{R}^2 \mid p_2 \geq p_1 + z \text{ and } q_2 \geq q_1 + z, \text{ and } 0 \leq p_1, p_2, q_1, q_2 \leq 3z\}.$$

It is not hard to compute its volume $4z^4$ which increases with $z$, but its primal counterpart contracts and converges to a single point $((0, 1), (0, 1))$.

We also note that (B) remains true in the dual space used by Eshel and Akin. An example is

$$S(z) = \{((p_1 - p_3, p_2 - p_3), (q_1 - q_3, q_2 - q_3)) \mid z \leq p_1 - p_3, q_1 - q_3 \leq 2z \text{ and } -2z \leq p_2 - p_3, q_2 - q_3 \leq -z\}.$$

The volume of $S(z)$ is $z^4$ which increases with $z$, but its primal counterpart converges to the primal point $((1, 0, 0), (1, 0, 0))$ as $z \to \infty$.

Proposition 1 follows directly from the following proposition.

**Proposition 9.** *Let $S$ be a set in the dual space with Lebesgue volume $v$. Also, suppose there exists $j \in S_1$ and $k \in S_2$ such that $\max_{(\mathbf{p},\mathbf{q}) \in S} p_j - \min_{(\mathbf{p},\mathbf{q}) \in S} p_j \leq R_j$ and $\max_{(\mathbf{p},\mathbf{q}) \in S} q_k - \min_{(\mathbf{p},\mathbf{q}) \in S} q_k \leq R_k$. Then for $\beta := \exp\left( \left( \frac{v}{R_j R_k} \right)^{1/(n+m-2)} \right)$, at least one of the followings holds:*

- *There exists $j' \in S_1$ such that $\left( \max_{(\mathbf{p},\mathbf{q}) \in S} \frac{x_{j'}(\mathbf{p})}{x_j(\mathbf{p})} \right) \Big/ \left( \min_{(\mathbf{p},\mathbf{q}) \in S} \frac{x_{j'}(\mathbf{p})}{x_j(\mathbf{p})} \right) \geq \beta$. Furthermore, if there exists $(\mathbf{p}^\#, \mathbf{q}^\#) \in S$ such that $x_j(\mathbf{p}^\#), x_{j'}(\mathbf{p}^\#) \geq \kappa > 0$, then the diameter of $\mathsf{G}(S)$ w.r.t. $\ell_2$ norm is at least $\left( 1 - \beta^{-1/4} \right) \kappa$.*

- *There exists $k' \in S_2$ such that $\left( \max_{(\mathbf{p},\mathbf{q}) \in S} \frac{y_{k'}(\mathbf{q})}{y_k(\mathbf{q})} \right) \Big/ \left( \min_{(\mathbf{p},\mathbf{q}) \in S} \frac{y_{k'}(\mathbf{q})}{y_k(\mathbf{q})} \right) \geq \beta$. Furthermore, if there exists $(\mathbf{p}^\#, \mathbf{q}^\#) \in S$ such that $y_k(\mathbf{q}^\#), y_{k'}(\mathbf{q}^\#) \geq \kappa > 0$, then the diameter of $\mathsf{G}(S)$ w.r.t. $\ell_2$ norm is at least $\left( 1 - \beta^{-1/4} \right) \kappa$.*

*Proof.* Without loss of generality, we assume that $j = 1$ and $k = 1$. Consider the mapping:

$$((p_1, p_2, \cdots, p_n), (q_1, q_2, \cdots, q_m)) \rightarrow ((p_1, p_2 - p_1, \cdots, p_n - p_1), (q_1, q_2 - q_1, \cdots, q_m - q_1)).$$

This is a linear mapping, and it is easy to verify that the determinant of the matrix that describes this linear mapping has determinant 1, so the mapping is volume-preserving.

Suppose that each of the quantities $p_{j'} - p_1$ and $q_{k'} - q_1$ is bounded by an interval of length at most $R$ within the set $S$, for a value of $R$ to be specified later. Then $S$ is a subset of a rectangular box in $\mathbb{R}^{n+m}$, with $n + m - 2$ sides of lengths at most $R$, and the remaining two sides of lengths at most $R_j$ and $R_k$. Thus, the volume of $S$ after the above linear mapping is at most $R^{n+m-2} R_j R_k$. When $R < \left( \frac{v}{R_j R_k} \right)^{1/(n+m-2)}$, this is a contradiction.

Thus, there exists one quantity $p_{j'} - p_1$ or $q_{k'} - q_1$ which is *not* bounded by an interval of length at most $\left( \frac{v}{R_j R_k} \right)^{1/(n+m-2)}$. Then we are done by recalling that $\frac{x_{j'}(\mathbf{p})}{x_1(\mathbf{p})} = \exp(p_{j'} - p_1)$ and $\frac{y_{k'}(\mathbf{q})}{y_1(\mathbf{q})} = \exp(q_{k'} - q_1)$.

If furthermore, there exists $(\mathbf{p}^\#, \mathbf{q}^\#) \in S$ such that $x_j(\mathbf{p}^\#), x_{j'}(\mathbf{p}^\#) \geq \kappa > 0$, then there exists $(\mathbf{p}^*, \mathbf{q}^*) \in S$ such that either

$$\frac{x_j(\mathbf{p}^*)}{x_{j'}(\mathbf{p}^*)} \Big/ \frac{x_j(\mathbf{p}^\#)}{x_{j'}(\mathbf{p}^\#)} \geq \beta^{1/2} \quad \text{or} \quad \frac{x_j(\mathbf{p}^*)}{x_{j'}(\mathbf{p}^*)} \Big/ \frac{x_j(\mathbf{p}^\#)}{x_{j'}(\mathbf{p}^\#)} \leq \beta^{-1/2}.$$

We focus on the former case, as the latter case is similar. We have $x_j(\mathbf{p}^*) - x_j(\mathbf{p}^\#) \geq x_j(\mathbf{p}^\#) \cdot \left( \frac{x_{j'}(\mathbf{p}^*)}{x_{j'}(\mathbf{p}^\#)} \cdot \beta^{1/2} - 1 \right)$. If $\frac{x_{j'}(\mathbf{p}^*)}{x_{j'}(\mathbf{p}^\#)} \geq \beta^{-1/4}$, we have $x_j(\mathbf{p}^*) - x_j(\mathbf{p}^\#) \geq \kappa(\beta^{1/4} - 1) \geq \kappa(1 - \beta^{-1/4})$. Otherwise, $\frac{x_{j'}(\mathbf{p}^*)}{x_{j'}(\mathbf{p}^\#)} < \beta^{-1/4}$, and hence $x_{j'}(\mathbf{p}^\#) - x_{j'}(\mathbf{p}^*) > x_{j'}(\mathbf{p}^\#) \cdot \left( 1 - \beta^{-1/4} \right) \geq \kappa(1 - \beta^{-1/4})$. $\qquad\square$

# B  Unavoidability of MWU in Games

*Proof of Lemma 4.* We suppose the contrary, i.e., for all $\tau \leq T$, $S(\tau) \subset V$, where $T$ will be specified later. We analyze how the volume of $S(t)$ changes with $t$ using formula (6). We rewrite it here:

$$\mathsf{vol}(S(t+1)) = \int_{(\mathbf{p},\mathbf{q}) \in S(t)} \det (\mathbf{I} + \epsilon \cdot \mathbf{J}(\mathbf{p}, \mathbf{q})) \, dV.$$

By Lemma 2, if $S(t) \subset V$, then the above inequality yields $\mathsf{vol}(S(t+1)) \geq \mathsf{vol}(S(t)) \cdot \left( 1 + (\overline{C} - \epsilon)\epsilon^2 \right)$, and hence

$$\forall t \leq T + 1, \quad \mathsf{vol}(S(t)) \geq \mathsf{vol}(S) \cdot \left( 1 + (\overline{C} - \epsilon)\epsilon^2 \right)^t. \tag{17}$$

On the other hand, observe that in the update rule (3), each variable is changed by a value in the interval $[-\epsilon, \epsilon]$ per time step, since every entry in $\mathbf{A}, \mathbf{B}$ is in the interval $[-1, 1]$. Consequently, the

range of possible values for each variable in $S(t)$ lies within an interval of length at most $d(S) + 2\epsilon t$, and hence $S(t)$ is a subset of a hypercube with side length $d(S) + 2\epsilon t$. Therefore,

$$\forall t \leq T + 1, \quad \text{vol}(S(t)) \ \leq \ (d(S) + 2\epsilon t)^{n+m}. \tag{18}$$

Note that the lower bound in (17) is exponential in $t$, while the upper bound in (18) is polynomial in $t$. Intuitively, it is clear that the two bounds cannot be compatible for some large enough $T$, and hence a contradiction. The rest of this proof is to derive how large $T$ should be. Precisely, we seek $T$ such that

$$(d(S) + 2\epsilon T)^{n+m} \ < \ \text{vol}(S) \cdot \left(1 + (\overline{C} - \epsilon)\epsilon^2\right)^T.$$

First, we impose that $T \geq d(S)/(2\epsilon) =: T_1$. Taking logarithm on both sides, to satisfy the above inequality, it suffices that

$$(n+m)\ln(4\epsilon T) \ < \ \frac{T \cdot (\overline{C} - \epsilon)\epsilon^2}{2} + \ln(\text{vol}(S)).$$

Since $4\epsilon \leq 1$, it suffices that

$$(\overline{C} - \epsilon)\epsilon^2 T - 2(n+m)\ln T \ > \ 2 \cdot \ln \frac{1}{\text{vol}(S)}.$$

Next, observe that when $T \geq \frac{8(n+m)}{(\overline{C}-\epsilon)\epsilon^2} \ln \frac{4(n+m)}{(\overline{C}-\epsilon)\epsilon^2} =: T_2$, we have $(\overline{C} - \epsilon)\epsilon^2 T - 2(n+m)\ln T \geq (\overline{C} - \epsilon)\epsilon^2 T/2$. (We will explain why in the next paragraph.) Then it is easy to see that $T \geq \frac{4}{(\overline{C}-\epsilon)\epsilon^2} \ln \frac{1}{\text{vol}(S)} =: T_3$ suffices. Overall, we need $T = \max\{T_1, T_2, T_3\}$.

Lastly, we explain why the inequality in the last paragraph holds. Observe that it is equivalent to $\frac{T}{\ln T} \geq \frac{4(n+m)}{(\overline{C}-\epsilon)\epsilon^2} =: \gamma$. Then it suffices to know that $\frac{T}{\ln T}$ is an increasing function of $T$ when $T \geq 3$, and

$$\frac{T_2}{\ln T_2} \ = \ \frac{2\gamma \ln \gamma}{\ln 2 + \ln \gamma + \ln \ln \gamma} \ \geq \ \frac{2\gamma \ln \gamma}{2\ln \gamma} \ = \ \gamma,$$

where the only inequality sign in the above expression holds because $\ln \gamma \geq \ln \ln \gamma + \ln 2 > 0$ when $\gamma \geq 3$. □

*Proof of Theorem 3.* Let $U'$ denote the set of points in $U$ which, when taken as a starting point, will eventually reach a point outside $U$. Suppose the theorem does not hold, i.e., $U'$ is not dense. Then we can find a primal open set $B \subset U$ such that its flow must stay in $U$ forever.

Let $V := \mathsf{G}^{-1}(U)$, $S' := \mathsf{G}^{-1}(B)$. Due to the discussion immediately after (7) and the assumption that $U$ is uncontrollable in the primal space, $V$ is uncontrollable in the dual space. On the other hand, $S'$ is open and unbounded; but it is easy to find a subset $S \subset S'$ which is open and bounded. Thus, $S$ has positive and finite volume. We apply Lemma 4 with the sets $V, S$ given above, to show that $\Phi(\tau, S)$ at some time $\tau$ contains a point $(\mathbf{p}^\tau, \mathbf{q}^\tau) \notin V$. By definition of $V$, $\mathsf{G}(\mathbf{p}^\tau, \mathbf{q}^\tau) \notin U$.

Let $(\mathbf{p}^0, \mathbf{q}^0)$ denote a point in $S$ such that its flow at time $\tau$ is $(\mathbf{p}^\tau, \mathbf{q}^\tau)$. Since $S$ is a subset of $S'$, $\mathsf{G}(\mathbf{p}^0, \mathbf{q}^0) \in B$. Due to the equivalence between the primal update rule (4) and the dual update rule (3), we can conclude that when $\mathsf{G}(\mathbf{p}^0, \mathbf{q}^0) \in B$ is used as the starting point of the primal update rule (4), at time $\tau$ its flow is $\mathsf{G}(\mathbf{p}^\tau, \mathbf{q}^\tau)$ which is not in $U$, a contradiction. □

## C  Extremism of MWU in Zero-Sum Games

**Lemma 10.** *Suppose an agent has $m$ options which she use MWU with step-size $\epsilon$ to decide the mixed strategy $\mathbf{y}^t = (y_1^t, \cdots, y_m^t)$ in each time step. Suppose at each round $t$, the payoff to each option $k$ is $a_k + \delta_k^t$, where*

- *each $a_k \in [-1, 1]$;*
- *there exists a positive number $\alpha_2 > 0$, such that for any $2 \leq k \leq m$, $a_{k-1} - a_k \geq \alpha_2$;*
- *there exists a positive number $\delta \leq \alpha_2/8$, such that $\delta_k^t \in [-2\delta, 2\delta]$.*

*Let $\hat{k}(t)$ denote the strategy $\min\{k \in [m] \mid y_k^t > \delta/(m-1)\}$. Then for $T := \left\lceil \frac{2}{\epsilon(\alpha_2 - 4\delta)} \cdot \ln \frac{m-1}{\delta} \right\rceil$, (i) if $\mathbf{y}^{\tau+T}$ has more than one entries larger than $\delta/(m-1)$ for some $\tau \geq 0$, then $\hat{k}(\tau + T) \leq \hat{k}(\tau) - 1$, and (ii) for some $t \leq (m-1)T$, $\mathbf{y}^t$ has an entry which is larger than or equal to $1 - \delta$.*

*Proof.* For part (i), we prove the contrapositive statement instead: if $\hat{k}(\tau + T) \geq \hat{k}(\tau)$, then $\mathbf{y}^{\tau+T}$ has exactly one entry larger than $\delta/(m-1)$.

Let $k = \hat{k}(\tau)$. For any $\ell > k$, due to the definition of the MWU update rule (4) and our assumptions, for $t \geq \tau$,

$$\frac{y_\ell^{t+1}}{y_k^{t+1}} = \frac{y_\ell^t}{y_k^t} \cdot \exp\left(\epsilon(a_\ell + \delta_\ell^t - a_k - \delta_k^t)\right) \leq \frac{y_\ell^t}{y_k^t} \cdot \exp\left(-\epsilon(\alpha_2 - 4\delta)\right).$$

Since $k = \hat{k}(\tau)$, we have $y_k^\tau > \delta/(m-1)$. Also, $y_k^t, y_\ell^\tau \leq 1$ trivially. Thus, for any $t \geq \tau$,

$$y_\ell^t \leq y_k^t \cdot \frac{y_\ell^\tau}{y_k^\tau} \cdot \exp\left(-\epsilon(\alpha_2 - 4\delta)(t - \tau)\right) < \frac{m-1}{\delta} \cdot \exp\left(-\epsilon(\alpha_2 - 4\delta)(t - \tau)\right).$$

When $\exp\left(-\epsilon(\alpha_2 - 4\delta)(t - \tau)\right) \leq \delta^2/(m-1)^2$, or equivalently $t \geq \tau + \left\lceil \frac{2}{\epsilon(\alpha_2 - 4\delta)} \cdot \ln \frac{m-1}{\delta} \right\rceil = \tau + T$, we have $y_\ell^t \leq \delta/(m-1)$.

Due to the conclusion of the last paragraph, we have $\hat{k}(\tau + T) \leq k$. But we also have the assumption $\hat{k}(\tau+T) \geq \hat{k}(\tau) = k$. Thus, $\hat{k}(\tau+T) = k$, and hence for any $k' < k$, $y_{k'}^{\tau+T} \leq \delta/(m-1)$. This, together with the conclusion of the last paragraph, shows that $y_k^{\tau+T}$ is the only entry in $\mathbf{y}^{\tau+T}$ which is larger than $\delta/(m-1)$. This completes the proof of part (i).

We prove part (ii) by contradiction. Suppose that for all $t \leq (m-1)T$, $\mathbf{y}^t$ has more than one entries larger than $\delta/(m-1)$. First of all, $\hat{k}(0) \neq m$, for otherwise $y_m^0$ is the only entry in $\mathbf{y}^0$ which is larger than $\delta/(m-1)$. Next, we apply part (i) for $(m-1)$ times to yield that $\hat{k}((m-1)T) \leq \hat{k}(0) - (m-1) \leq 0$, a contradiction. Thus, for some $\mathbf{y}^t$ with $t \leq (m-1)T$, it has exactly one entry which is larger than $\delta/(m-1)$. The entry has to be larger than or equal to $1 - (m-1)(\delta/(m-1)) = 1 - \delta$. $\qquad\square$

Let $\mathcal{E}_{a,b}^\delta$ be the collection of all points $(\mathbf{x}, \mathbf{y})$, such that at least $a$ entries in $\mathbf{x}$ are larger than $\delta$, and at least $b$ entries in $\mathbf{y}$ are larger than $\delta$.

*Proof of Theorem 5.* The proof comprises of three steps.

**Step 1.** We show that for any $\kappa > 0$, $\mathcal{E}_{2,2}^\kappa$ is an uncontrollable primal set with

$$\inf_{(\mathbf{x},\mathbf{y}) \in \mathcal{E}_{2,2}^\kappa} C(\mathbf{x}, \mathbf{y}) \geq \kappa^2(\alpha_1)^2/2. \tag{19}$$

Recall Lemma 2 that $C(\mathbf{x}, \mathbf{y})$ is the variance of a random variable $X$, which is equal to $\mathbb{E}\left[(X - \mathbb{E}[X])^2\right]$. For any point $(\mathbf{x}, \mathbf{y}) \in \mathcal{E}_{2,2}^\kappa$, each of $\mathbf{x}, \mathbf{y}$ has at least two entries larger than $\kappa$. Suppose $x_{j_1}, x_{j_2}, y_{k_1}, y_{k_2} > \kappa$. Then

$$C(\mathbf{x}, \mathbf{y}) \geq \sum_{j \in \{j_1, j_2\}} \sum_{k \in \{k_1, k_2\}} \kappa^2 \left[\underbrace{\left(A_{jk} - [\mathbf{A}\mathbf{y}]_j - [\mathbf{A}^\mathsf{T}\mathbf{x}]_k\right) - \mathbb{E}[X]}_{A'_{jk}}\right]^2. \tag{20}$$

Due to Condition (A) and Equation (1), we are guaranteed that among the four possible values of $A'_{jk}$, the maximum and minimum values differ by at least $\alpha_1$, for otherwise we can choose $a_j = [\mathbf{A}\mathbf{y}]_j + \mathbb{E}[X]$ and $b_k = -[\mathbf{A}^\mathsf{T}\mathbf{x}]_k$ in (1) to show that the $2 \times 2$ sub-matrix of $\mathbf{A}$ corresponding to strategies $\{j_1, j_2\} \times \{k_1, k_2\}$ has distance from triviality strictly less than $\alpha_1$. Consequently, $C(\mathbf{x}, \mathbf{y}) \geq \kappa^2(\alpha_1/2)^2 \cdot 2 = \kappa^2(\alpha_1)^2/2$.

**Step 2.** Then we apply Theorem 3 to show that for any step-size $\epsilon < \min\left\{\frac{1}{32n^2m^2}, \frac{\kappa^2(\alpha_1)^2}{2}\right\}$, there exists a dense subset of points in $\mathsf{int}(\Delta)$ such that the flow of each such point must eventually reach a point outside $\mathcal{E}_{2,2}^\kappa$. Let $(\hat{\mathbf{x}}, \hat{\mathbf{y}})$ denote the point outside $\mathcal{E}_{2,2}^\kappa$. At $(\hat{\mathbf{x}}, \hat{\mathbf{y}})$, one of the two players, which we assume to be Player 1 without loss of generality, concentrates her game-play on only one strategy, which we denote by strategy $\hat{j}$. Precisely, for any $j \neq \hat{j}$, $\hat{x}_j \leq \kappa$, and hence $\sum_{j \in S_1 \setminus \{\hat{j}\}} \hat{x}_j \leq (N-1)\kappa$.

**Step 3.** Now, we consider the flow starting from $(\hat{\mathbf{x}}, \hat{\mathbf{y}})$. Since $x_j^{t+1}/x_j^t \le \exp(2\epsilon)$ always, we are sure that for the next $T_1 := \left\lfloor \frac{1}{2\epsilon} \ln \frac{\delta}{(N-1)\kappa} \right\rfloor$ time steps, $\sum_{j \in S_1 \setminus \{\hat{j}\}} x_j^t \le \delta$. Thus, within this time period, the payoff to strategy $k$ of Player 2 in each time step is $-A_{\hat{j}k}$ plus a perturbation term in the interval $[-2\delta, 2\delta]$. Then by Lemma 10 part (ii) (a sanity check on the conditions required by the lemma is easy and thus skipped), if $(N-1) \cdot \left\lceil \frac{2}{\epsilon(\alpha_2 - 4\delta)} \cdot \ln \frac{N-1}{\delta} \right\rceil \le T_1$, we are done. A direct arithmetic shows that this inequality holds if $\kappa \le (\delta/(N-1))^{4(N-1)/(\alpha_2 - 4\delta)+1}/3$.  $\square$

*Proof of Theorem 6.* By Theorem 5, we are guaranteed that there exists a dense subset of starting points such that the flow of each of them must eventually reach the extremal domain with threshold $\delta$. When we apply Theorem 5, This is our starting point to prove Theorem 6.

**Step 1.** We show that: for each such starting point $y$, we prove that its flow cannot remain in the extremal domain forever.

First, observe that the extremal domain is the union of small neighbourhoods of extremal points, and each such neighbourhood is far from the other neighbourhoods.

Suppose the contrary that there exists a starting point such that its flow remains in the extremal domain forever. Due to the above observation, its flow must remain in the small neighbourhood of *one* extremal point forever. Suppose the utility values at this extremal point is $(u, -u)$; recall that by assumption, $|u - v| \ge r$. Since the flow remains near this extremal point, in the long run, the average utility gained by Player 1 must lie in the interval $(1 - \delta)u \pm \delta$, which is a subset of the interval $u \pm 2\delta$.

On the other hand, due to a well-known regret bound of MWU (see, for instance, [10, Lemma 9]), in the long run, the average utility gained by Player 1 must lie in the interval $v \pm 3\epsilon$. When $3\epsilon + 2\delta \le r/2$, this is incompatible with the interval derived in the previous paragraph, thus a contradiction.

**Step 2.** Indeed, we have a stronger version of the result in Step 1. Recall that the complement of the extremal domain is an open set. Since the MWU update rule is a continuous mapping, it preserves openness, and hence we not only one point $y$ that visits and leaves the extremal domain, but we have an open neighbourhood $\mathcal{O}_1$ around $y$, such that the flow of $\mathcal{O}_1$ visits and leaves the extremal domain. Let $\mathcal{O}'$ denote the flow of $\mathcal{O}_1$ at the moment when the flow leaves the extremal domain. $\mathcal{O}'$ is open, and hence has positive Lebesgue measure.

Then we construct a closed subset $\mathcal{C}_1 \subset \mathcal{O}_1$ with positive Lebesgue measure. This is easy as follows. First, we take an arbitrary point $z \in \mathcal{O}'$. Since $\mathcal{O}'$ is open, there exists an open ball around $z$ with some radius $r > 0$ which is contained in $\mathcal{O}'$. Since the MWU update rule is a continuous mapping, its inverse for arbitrary finite time preserves closeness, the inverse (back to the starting time) of the closed ball around $z$ with radius $r/2$ is a closed set, which we take as $\mathcal{C}_1$; $\mathcal{C}_1 \subset \mathcal{O}_1$ since the closed ball around $z$ with radius $r/2$ is a subset of $\mathcal{O}'$, and the inverse (back to the starting time) of $\mathcal{O}'$ is $\mathcal{O}_1$.

**Step 3.** Since $\mathcal{C}_1$ has positive Lebesgue measure, we can reiterate the arguments in Steps 1 and 2, and construct open set $\mathcal{O}_2 \subset \mathcal{C}_1$ and closed set $\mathcal{C}_2 \subset \mathcal{O}_2$ that visit and leave the extremal domain again.

By iterating these arguments repeatedly, we get a sequence of closed (and indeed compact) sets $\mathcal{C}_1 \supset \mathcal{C}_2 \supset \mathcal{C}_3 \supset \cdots$. By the Cantor's intersection theorem, the intersection of this sequence of closed sets must be non-empty. Then any point in this intersection is a starting point that visits and leaves the extremal domain infinitely often.  $\square$

### C.1 Classical Rock-Paper-Scissors Game

The standard Rock-Paper-Scissors game is the zero-sum game $(\mathbf{A}, -\mathbf{A})$ with the following payoff matrix: $\mathbf{A} = \begin{bmatrix} 0 & -1 & 1 \\ 1 & 0 & -1 \\ -1 & 1 & 0 \end{bmatrix}$. There are two types of $2 \times 2$ sub-matrices of $\mathbf{A}$. Consider such a sub-matrix which corresponds to strategy set $Q_i \subset \{R, P, S\}$ for Players $i = 1, 2$. The first type is when $Q_1 = Q_2$, then the sub-matrix is $\mathbf{A}' = \begin{bmatrix} 0 & -1 \\ 1 & 0 \end{bmatrix}$, which is trivial, i.e., $c(\mathbf{A}') = 0$. The second type is when $|Q_1 \cap Q_2| = 1$, then the sub-matrix is $\mathbf{A}'' = \begin{bmatrix} 0 & 1 \\ 1 & -1 \end{bmatrix}$; it is easy to show that $c(\mathbf{A}'') = 3/2$. Due to the existence of the first type of sub-matrices, Theorem 5 cannot be applied. We provide a separate proof to show that the same conclusion of Theorem 5 holds for this specific game.

**Theorem 11.** *Suppose the underlying game is the standard Rock-Paper-Scissors game. For any $0 < \delta < 1/20$, if both players use MWU with step-size $\epsilon$ satisfying $\epsilon < \delta^{22}/(34 \times 10^6)$, then there exists a dense subset of points in $\mathrm{int}(\Delta)$, such that the flow of each such point must eventually reach a point $(\mathbf{x}, \mathbf{y})$ where each of $\mathbf{x}, \mathbf{y}$ has exactly one entry larger than or equal to $1 - \delta$.*

*Proof.* To start, we define a new family of primal set $\mathcal{E}^\kappa$. To define it, let $(\mathbf{x}, \mathbf{y})$ be a point in $\mathrm{int}(\Delta)$, and let $Q_i$ denote the set of strategies of Player 1 with probability density larger than $\kappa$. Then $(\mathbf{x}, \mathbf{y}) \in \mathcal{E}^\kappa$ if and only if $|Q_1|, |Q_2| \geq 2$, and furthermore, there exists $Q_1' \subset Q_1$, $Q_2' \subset Q_2$ such that $|Q_1'|, |Q_2'| = 2$ and $|Q_1' \cap Q_2'| = 1$.

The definition of $\mathcal{E}^\kappa$ deliberately avoids us from deriving a lower bound of $C(\mathbf{x}, \mathbf{y})$ in the manner of (20) when $\{j_1, j_2\} = \{k_1, k_2\}$, which corresponds to a trivial sub-matrix. Then by following Step 1 in the proof of Theorem 5, we have $\inf_{\mathbf{x}, \mathbf{y} \in \mathcal{E}^\kappa} \geq \kappa^2 c(\mathbf{A}'')^2/2 = 9\kappa^2/8$. By following Step 2 in the proof of Theorem 5, when $\epsilon < \min\left\{\frac{1}{32n^2m^2}, \frac{9\kappa^2}{8}\right\}$, there exists a dense subset of points in $\mathrm{int}(\Delta)$ such that the flow of each such point must reach a point $(\hat{\mathbf{x}}, \hat{\mathbf{y}})$ outside $\mathcal{E}^\kappa$.

Below, we assume the time is reset to zero with starting point $(\hat{\mathbf{x}}, \hat{\mathbf{y}})$. We proceed on a case analysis below.

**Case 1: either $|Q_1| = 1$ or $|Q_2| = 1$.** For this case, we can simply follow Step 3 in the proof of Theorem 5. $\kappa \leq \delta^{11}/6144$ suffices.

**Case 2: $Q_1 = Q_2$, and $|Q_1| = 2$.** Without loss of generality, we assume $Q_1 = Q_2 = \{R, P\}$. In the sub-game corresponding to $Q_1 \times Q_2$, each player has a strictly dominant strategy, namely $P$. Intuitively, the probability of choosing strategy $P$ must strictly increase with time (when we ignore the tiny effect of strategy $S$).

More formally, starting from time zero, for the next $T_1 := \left\lfloor \frac{1}{2\epsilon} \ln \frac{\delta}{2\kappa} \right\rfloor$ time steps, $x_S^t, y_S^t \leq \delta/2$, and hence $x_P^t + x_R^t, y_P^t + y_R^t \geq 1 - \delta/2$. Then

(the payoff to strategy $P$ of Player 1 in round $t$) $-$ (the payoff to strategy $R$ of Player 1 in round $t$)

$$= \left[y_P^t \cdot 0 + y_R^t \cdot 1 + y_S^t \cdot (-1)\right] - \left[y_P^t \cdot (-1) + y_R^t \cdot 0 + y_S^t \cdot 1\right]$$

$$\geq y_P^t + y_R^t - \delta \geq 1 - 2\delta.$$

Thus, $\frac{x_P^{t+1}}{x_R^{t+1}} \geq \frac{x_P^t}{x_R^t} \cdot \exp\left(\epsilon(1 - 2\delta)\right)$, and hence

$$\frac{x_P^t}{x_R^t} \geq \hat{x}_P \cdot \exp\left(\epsilon(1 - 2\delta)t\right). \tag{21}$$

The above inequality holds also when all $x$'s are replaced by $y$'s.

- **Case 2(a): at $(\hat{\mathbf{x}}, \hat{\mathbf{y}})$, each of the two players have one strategy with probability larger than or equal to $1 - \delta$.** Then we are done.

- **Case 2(b): at $(\hat{\mathbf{x}}, \hat{\mathbf{y}})$, each of the two players have all strategies with probability less than $1 - \delta$.** Then we know that $\hat{x}_P, \hat{y}_P \geq 1 - (1 - \delta) - \delta/2 = \delta/2$. By (21), when $\exp\left(\epsilon(1 - 2\delta)t\right) \geq 4/\delta^2$, we have $x_P^t/x_R^t, y_P^t/y_R^t \geq 2/\delta$. And since we still have $x_S^t, y_S^t \leq \delta/2$, it is easy to show that $x_P^t, y_P^t \geq 1 - \delta$.

- **Case 2(c): at $(\hat{\mathbf{x}}, \hat{\mathbf{y}})$, exactly one of the two players have one strategy with probability larger than or equal to $1 - \delta$.** Without loss of generality, we assume the player is Player 2. Then we know that $\hat{x}_P, \hat{x}_R \geq \delta/2$. Similar to the argument for Case 2(b), when $\exp\left(\epsilon(1 - 2\delta)t\right) \geq 4/\delta^2$, we have $x_P^t \geq 1 - \delta$.

  If at this time $t$, we have either $y_P^t \geq 1 - \delta$ or $y_R^t \geq 1 - \delta$, we are done. Otherwise, we have $y_P^t \geq \delta/2$. Thus, after another period of time $t'$ such that $\exp\left(\epsilon(1 - 2\delta)t'\right) \geq 4/\delta^2$, we have $y_P^{t+t'} \geq 1 - \delta$, while $x_P^{t+t'} \geq 1 - \delta$ still.

For the arguments for Cases 2(b),(c) to hold, we need

$$2 \cdot \left\lceil \frac{1}{(1 - 2\delta)\epsilon} \ln \frac{4}{\delta^2} \right\rceil \leq T_1,$$

A direct arithmetic shows that $\kappa \leq \delta^{10}/2845$ suffices. $\qquad \square$

# D  Continuous Analogue of OMWU

## D.1  Some Alternative Approaches for Volume Analysis of OMWU, and Their Drawbacks

Recall that OMWU update rule (5) at time $t + 1$ depends on the past updates at times $t$ and $t - 1$, so we cannot apply (6) directly for its volume analysis.

At first sight, it might seem necessary to perform volume analysis in the product space $\Delta \times \Delta$ that contains $((\mathbf{p}_t, \mathbf{q}_t), (\mathbf{p}_{t-1}, \mathbf{q}_{t-1}))$. However, this raises a number of technical difficulties. First, since the initialization sets $(\mathbf{p}_1, \mathbf{q}_1)$ as a function of $(\mathbf{p}_0, \mathbf{q}_0)$, the initial set has to lie in a proper manifold in $\Delta \times \Delta$, thus it has zero Lebesgue measure w.r.t. $\Delta \times \Delta$, making volume analysis useless, as the volume must remain zero when the initial set is of measure zero. In some cases, it might be possible to define a useful *volume form* in a proper manifold, but the corresponding volume analysis will require some heavy machinery from calculus of manifold, which we do not see a simple way to implement.

Second, even if we permit $\mathbf{p}_1, \mathbf{q}_1$ to be unrelated to $\mathbf{p}_0, \mathbf{q}_0$ so that we can permit an initial set with positive measure, the OMWU update rule is not of the same type that is presumed by the formula (6). We will need to use the more general form of integration by substitution, and the volume integrand there will *not* be of the form $\mathbf{I} + \epsilon \cdot \mathbf{J}$, hence the determinant is not a polynomial of $\epsilon$ with constant term 1. We tried this approach, but were not able to make any meaningful observation from it.

## D.2  Informational Contexts and Constraints for Approximating $\dot{\mathbf{u}}(t)$

We list three relevant contexts for computing or approximating $\dot{\mathbf{u}}(t)$, which is then used in OMWU update rule. The last context is what we have used in this paper.

1. If the function $\mathbf{u}$ is explicitly given and it is a simple function of time (e.g. a polynomial), the function $\dot{\mathbf{u}}$ can be explicitly computed. Euler method with step-size $\Delta t = \epsilon$ is the update rule

$$\mathbf{p}(t + \epsilon) = \mathbf{p}(t) + \epsilon \cdot \mathbf{u}(t) + \epsilon^2 \cdot \dot{\mathbf{u}}(t).$$

2. However, in some scenarios, $\mathbf{u}$ is a rather complicated function of $t$, so computing explicit formula for $\dot{\mathbf{u}}$ might not be easy. Yet, we have full knowledge of values of $\mathbf{u}(0), \mathbf{u}(\Delta t), \mathbf{u}(2 \cdot \Delta t), \mathbf{u}(3 \cdot \Delta t), \cdots$. Then a common approach to approximately compute $\dot{\mathbf{u}}(N \cdot \Delta t)$ is to use the central finite-difference method:

$$\dot{\mathbf{u}}(N \cdot \Delta t) \;=\; \frac{\mathbf{u}((N + 1) \cdot \Delta t) - \mathbf{u}((N - 1) \cdot \Delta t)}{2 \cdot \Delta t} \;+\; \mathcal{O}((\Delta t)^2).$$

Euler method with step-size $\Delta t = \epsilon$ which makes use of the above approximation gives the update rule

$$\mathbf{p}(t + \epsilon) = \mathbf{p}(t) + \epsilon \cdot \mathbf{u}(t) + \epsilon \cdot \frac{\mathbf{u}(t + \epsilon) - \mathbf{u}(t - \epsilon)}{2}.$$

3. Even worse, in the context of online learning or game dynamics, at time $N \cdot \Delta t$, the players have only observed $\mathbf{u}(0), \mathbf{u}(\Delta t), \mathbf{u}(2 \cdot \Delta t), \cdots, \mathbf{u}(N \cdot \Delta t)$, but they do not have any knowledge on the *future* values of $\mathbf{u}$. Due to the more severe constraint on information, we have to settle with the backward finite-difference method to approximately compute $\dot{\mathbf{u}}(N \cdot \Delta t)$:

$$\dot{\mathbf{u}}(N \cdot \Delta t) \;=\; \frac{\mathbf{u}(N \cdot \Delta t) - \mathbf{u}((N - 1) \cdot \Delta t)}{\Delta t} \;+\; \mathcal{O}(\Delta t),$$

which has a higher-order error when compared with the central finite-difference method. Euler method with step-size $\Delta t = \epsilon$ which makes use of the above approximation gives the rule (9), by identifying $\mathbf{p}(t + \epsilon)$ as $\mathbf{p}^{t+1}$. Due to an error that occurs when we approximate $\dot{\mathbf{u}}$ as above,

$$\epsilon \cdot \mathbf{u}(t) + \epsilon \cdot (\mathbf{u}(t) - \mathbf{u}(t - 1)) \;=\; \epsilon \left[ \mathbf{u}(t) + \epsilon \cdot \dot{\mathbf{u}}(t) \right] + \mathcal{O}(\epsilon^3),$$

where the LHS is the quantity $\mathbf{p}^{t+1} - \mathbf{p}^t$ in the OMWU update rule (9), and the first term in the RHS is the standard Euler discretization of (10).

## D.3 From Recurrence of ODE to Standard ODE

In Equations (12) and (13), observe that each $\frac{dp_j}{dt}$ is expressed as an affine combination of various $\frac{dq_k}{dt}$, while each $\frac{dq_k}{dt}$ is expressed as an affine combination of various $\frac{dp_j}{dt}$. Thus, we may rewrite all these expressions into a matrix-form differential equation. Let $\mathbf{v}(\mathbf{p}, \mathbf{q})$ denote the following vector in $\mathbb{R}^{n+m}$:

$$\mathbf{v}(\mathbf{p}, \mathbf{q}) = ([\mathbf{A} \cdot \mathbf{y}(\mathbf{q})]_1, \cdots, [\mathbf{A} \cdot \mathbf{y}(\mathbf{q})]_n, \ [\mathbf{B}^\mathsf{T} \cdot \mathbf{x}(\mathbf{p})]_1, \cdots, [\mathbf{B}^\mathsf{T} \cdot \mathbf{x}(\mathbf{p})]_m)^\mathsf{T},$$

and let $\mathbf{M}(\mathbf{p}, \mathbf{q})$ denote the $(S_1 \cup S_2) \times (S_1 \cup S_2)$ matrix $\begin{bmatrix} \mathbf{0} & \mathbf{M}^1 \\ \mathbf{M}^2 & \mathbf{0} \end{bmatrix}$, where $\mathbf{M}^1 \equiv \mathbf{M}^1(\mathbf{p}, \mathbf{q})$ is a $S_1 \times S_2$ sub-matrix and $\mathbf{M}^2 \equiv \mathbf{M}^2(\mathbf{p}, \mathbf{q})$ is a $S_2 \times S_1$ sub-matrix defined as below:

$$M^1_{jk} = y_k(\mathbf{q}) \cdot (A_{jk} - [\mathbf{A} \cdot \mathbf{y}(\mathbf{q})]_j) \quad \text{and} \quad M^2_{kj} = x_j(\mathbf{p}) \cdot (B_{jk} - [\mathbf{B}^\mathsf{T} \cdot \mathbf{x}(\mathbf{p})]_k).$$

Then we can rewrite the recurrence system (12) and (13) as $\left( \frac{d\mathbf{p}}{dt}, \frac{d\mathbf{q}}{dt} \right)^\mathsf{T} = \mathbf{v}(\mathbf{p}, \mathbf{q}) + \epsilon \cdot \mathbf{M}(\mathbf{p}, \mathbf{q}) \cdot \left( \frac{d\mathbf{p}}{dt}, \frac{d\mathbf{q}}{dt} \right)^\mathsf{T}$. This can be easily solved to a standard (non-recurring) system of ODE:

$$\left( \frac{d\mathbf{p}}{dt}, \frac{d\mathbf{q}}{dt} \right)^\mathsf{T} = (\mathbf{I} - \epsilon \cdot \mathbf{M}(\mathbf{p}, \mathbf{q}))^{-1} \cdot \mathbf{v}(\mathbf{p}, \mathbf{q}),$$

if the inverse of the matrix $(\mathbf{I} - \epsilon \cdot \mathbf{M}(\mathbf{p}, \mathbf{q}))$ exists.

We proceed by using the following identity: if a square matrix $\mathbf{R}$ satisfies $\sup_{\|\mathbf{z}\|=1} \|\mathbf{R}\mathbf{z}\| < 1$, then $(\mathbf{I} - \mathbf{R})^{-1} = \mathbf{I} + \sum_{\ell=1}^\infty \mathbf{R}^\ell$. In our case, we desire $\sup_{\|\mathbf{z}\|=1} \|\epsilon \cdot \mathbf{M}(\mathbf{p}, \mathbf{q}) \cdot \mathbf{z}\| < 1$. Observe that for each row of $\mathbf{M}(\mathbf{p}, \mathbf{q})$, its $\ell_2$-norm is at most $2\|\mathbf{x}\|$ or $2\|\mathbf{y}\|$, which are upper bounded by 2. Thus, each entry in $\epsilon \cdot \mathbf{M}(\mathbf{p}, \mathbf{q}) \cdot \mathbf{z}$ is absolutely bounded by $2\epsilon$, and hence $\|\epsilon \cdot \mathbf{M}(\mathbf{p}, \mathbf{q}) \cdot \mathbf{z}\| \le 2\epsilon\sqrt{n+m}$. Consequently, $\epsilon < 1/(2\sqrt{n+m})$ suffices to guarantee that the inverse of $(\mathbf{I} - \epsilon \cdot \mathbf{M}(\mathbf{p}, \mathbf{q}))$ exists, and the identity mentioned above holds for its inverse:

$$\left( \frac{d\mathbf{p}}{dt}, \frac{d\mathbf{q}}{dt} \right)^\mathsf{T} = \left( \mathbf{I} + \sum_{\ell=1}^\infty \epsilon^\ell \cdot \mathbf{M}(\mathbf{p}, \mathbf{q})^\ell \right) \cdot \mathbf{v}(\mathbf{p}, \mathbf{q}).$$

## E  Volume Analysis of Discrete-Time OMWU

*Proof of Equation* (16). For the moment, we ignore the $\mathcal{O}(\epsilon^3)$ terms in (15). To use (6) for computing volume change, we need to derive $\epsilon \cdot \mathbf{J}(\mathbf{p}, \mathbf{q})$ in the volume integrand:

$$\forall j_1, j_2 \in S_1, \quad \epsilon J_{j_1 j_2} = \epsilon^2 \sum_{k \in S_2} y_k(\mathbf{q}) \cdot (A_{j_1 k} - [\mathbf{A} \cdot \mathbf{y}(\mathbf{q})]_{j_1}) \cdot x_{j_2}(\mathbf{p}) \cdot (B_{j_2 k} - [\mathbf{B}^\mathsf{T} \cdot \mathbf{x}(\mathbf{p})]_k);$$

$$\forall k_1, k_2 \in S_2, \quad \epsilon J_{k_1 k_2} = \epsilon^2 \sum_{j \in S_1} x_j(\mathbf{p}) \cdot (B_{jk_1} - [\mathbf{B}^\mathsf{T} \cdot \mathbf{x}(\mathbf{p})]_{k_1}) \cdot y_{k_2}(\mathbf{q}) \cdot (A_{jk_2} - [\mathbf{A} \cdot \mathbf{y}(\mathbf{q})]_j);$$

$$\forall j \in S_1, k \in S_2, \quad \epsilon J_{jk} = \epsilon \cdot y_k(\mathbf{q}) \cdot (A_{jk} - [\mathbf{A} \cdot \mathbf{y}(\mathbf{q})]_j) + \mathcal{O}(\epsilon^2);$$

$$\forall k \in S_2, j \in S_1, \quad \epsilon J_{kj} = \epsilon \cdot x_j(\mathbf{p}) \cdot (B_{jk} - [\mathbf{B}^\mathsf{T} \cdot \mathbf{x}(\mathbf{p})]_k) + \mathcal{O}(\epsilon^2).$$

With the above formulae, we expand $\det(\mathbf{I} + \epsilon \cdot \mathbf{J}(\mathbf{p}, \mathbf{q}))$ via the Leibniz formula. The determinant is of the form $1 + C'(\mathbf{p}, \mathbf{q}) \cdot \epsilon^2 + \mathcal{O}(\epsilon^3)$, where $C'(\mathbf{p}, \mathbf{q})$ is the coefficient of $\epsilon^2$ in the expression

$$\sum_{j \in S_1} \epsilon J_{jj} + \sum_{k \in S_2} \epsilon J_{kk} - \sum_{\substack{j \in S_1 \\ k \in S_2}} (\epsilon J_{jk})(\epsilon J_{kj}).$$

A straight-forward arithmetic shows the above expression equals to $-\epsilon^2 \cdot C_{(\mathbf{A}, \mathbf{B})}(\mathbf{p}, \mathbf{q}) + \mathcal{O}(\epsilon^3)$, and hence Equation (16) follows. $\qquad\square$

Recall from [11] that the volume integrand for MWU is

$$1 + C_{(\mathbf{A}, \mathbf{B})}(\mathbf{p}, \mathbf{q}) \cdot \epsilon^2 + \mathcal{O}(\epsilon^4),$$

while by (16), the volume integrand for OMWU is

$$1 - C_{(\mathbf{A},\mathbf{B})}(\mathbf{p},\mathbf{q}) \cdot \epsilon^2 + \mathcal{O}(\epsilon^3).$$

By (19), within $\mathsf{G}^{-1}(\mathcal{E}_{2,2}^\delta)$, $C_{(\mathbf{A},-\mathbf{A})}(\mathbf{p},\mathbf{q}) \geq \delta^2(\alpha_1)^2/2$, thus $C_{(\mathbf{A},\mathbf{A})}(\mathbf{p},\mathbf{q}) = -C_{(\mathbf{A},-\mathbf{A})}(\mathbf{p},\mathbf{q}) \leq -\delta^2(\alpha_1)^2/2$. Therefore, when $\epsilon$ is sufficiently small, the volume integrands for MWU in coordination game and OMWU in zero-sum game are both at most $1 - \epsilon^2\delta^2(\alpha_1)^2/4$.

**Corollary 12.** *Suppose the underlying game is a non-trivial zero-sum game* $(\mathbf{A}, -\mathbf{A})$ *and the parameter* $\alpha_1$ *as defined in Theorem 5 is strictly positive. For any* $1/2 > \delta > 0$, *for any sufficiently small* $0 < \epsilon \leq \bar{\epsilon}$ *where the upper bound depends on* $\delta$, *and for any set* $S = S(0) \subset \mathsf{G}^{-1}(\mathcal{E}_{2,2}^\delta)$ *in the dual space, if* $S$ *is evolved by the OMWU update rule* (5) *and if its flow remains a subset of* $\mathsf{G}^{-1}(\mathcal{E}_{2,2}^\delta)$ *for all* $t \leq T - 1$, *then* $\mathsf{vol}(S(T)) \leq \left(1 - \frac{\epsilon^2\delta^2(\alpha_1)^2}{4}\right)^T \cdot \mathsf{vol}(S).$

**Corollary 13.** *Suppose the underlying game is a non-trivial coordination game* $(\mathbf{A}, \mathbf{A})$ *and the parameter* $\alpha_1$ *as defined in Theorem 5 is strictly positive. For any* $1/2 > \delta > 0$, *for any sufficiently small* $0 < \epsilon \leq \bar{\epsilon}$ *where the upper bound depends on* $\delta$, *and for any set* $S = S(0) \subset \mathsf{G}^{-1}(\mathcal{E}_{2,2}^\delta)$ *in the dual space, if* $S$ *is evolved by the MWU update rule* (3) *and if its flow remains a subset of* $\mathsf{G}^{-1}(\mathcal{E}_{2,2}^\delta)$ *for all* $t \leq T - 1$, *then* $\mathsf{vol}(S(T)) \leq \left(1 - \frac{\epsilon^2\delta^2(\alpha_1)^2}{4}\right)^T \cdot \mathsf{vol}(S).$