[Reviews · NeurIPS 2020]

Review 1

Summary and Contributions: This paper provides volume analyses of MWU and OMWU, using new techniques from dynamical systems. The new analyses provide new insights into the two learning dynamics, going beyond just zero-sum games, to coordination games. These volume analyses established two novel while negative properties of MWU for zero-sum games, and also OMWU for coordination games: extremism and unavoidability. This complementary property of the two settings also implies some kind of "no-free-lunch theorem" in learning for games.

Strengths: The main strengths of the paper are the soundness of the theoretical claims, the novelty of the volume analyses for learning dynamics in games, and the significance of the results obtained via the new analyses. It is relevant to the community.

Weaknesses: The only small weakness might be a bit lack of empirical studies. The current empirical study only concerns two simple examples (though illustrative enough). It might be better to include more simulation examples, e.g., for general-sum games, or how the different choices of stepsize may affect the results, etc.

Correctness: The results seem mostly correct to me, though I did not check line-by-line. Empirical study is correct and clear.

Clarity: The paper is in general well-written. The logic is clear, and the idea is easy to follow. I enjoy reading the draft.

Relation to Prior Work: Mostly yes. It would be better to compare with the most relevant a prior work [9] with more details.

Reproducibility: Yes

Additional Feedback: Additional comments: 1. How may the results, particularly the volume analyses, be translated to the case with "diminishing stepsize"? What about other learning dynamics, e.g., those in [1][5][17]? 2. Typos: line 112, remove one "the"; the end of line 201, "following" -> "the following"; line 309 after system, need a "."; line 315, in "a" distributed manner; line 324, remove "the". ==================================== I have read the rebuttal, which has addressed my minor comments. Thanks for the clarifications.


Review 2

Summary and Contributions: The authors study continuous-time dynamical system analogues of multiplicative weights updates (MWU) and optimistic multiplicative weights updates (OMWU). They focus on zero-sum and coordination games, and use volume arguments to provide a theoretical analysis of the performance of MWU and OMWU in both of these settings. For zero-sum games, MWU exhibits exponential volume expansion and hence Lyapunov chaos with respect to the initial conditions of the learning dynamic, which gives theoretical evidence to numerical instability. On the other hand OMWU demonstrates volume contraction, which once again gives evidence towards the success of this dynamic. The authors also provide new measures of instability in equilibrium learning dynamics which they call extremism and unavoidability. Extremism is behaviour whereby a dynamic repeatedly reaches points nearby pure equilibria, irrespective of the set of equilibria (for example the game can have a unique mixed NE), and unavoidability is a setting whereby undesirable strategy profiles (under mild constraints) cannot be avoided indefinitely. With these definitions in hand, the authors demonstrate that for MWU in zero-sum games, the dynamic also suffers from extremism and unavoidability. OMWU on the other hand is not ideal in all scenarios, and the authors demonstrate that performance is essentially reversed in the worst case for coordination games. They show that MWU exhibits volume contraction and OMWU exhibits exponential volume expansion (hence Lyapunov instability), as well as extremism and unavoidability.

Strengths: These results seem to extend the existing literature on theoretical barriers to applying existing online learning algorithms to equilibrium computation. I am not as familiar with the techniques, but the work is relevant to the NeurIPS community, especially given increased interest in such decentralised equilibrium computation algorithms when applied to GAN training. The authors have also addressed reviewer concerns in the rebuttal phase well. My score remains the same.

Weaknesses: There could be more empirical results demonstrating the performance of MWU and OMWU side-by-side on given zero-sum and coordination games. The authors have also addressed this concern in the rebuttal phase well. My score remains the same.

Correctness: To my extent they look correct, though this is not my area of expertise.

Clarity: The paper is clear, which is a strong point, given the complicated machinery used.

Relation to Prior Work: This is done well.

Reproducibility: Yes

Additional Feedback:


Review 3

Summary and Contributions: This paper studies the unstability of MWU and OMWU on zero-sum games and coordination games respectively through volume analysis for the dynamic processes. The unstability is indicated by two properties, extremism (visiting near pure strategies infinitely) and unavoidability (escape any set of inital points eventually). These two negative propeties are novel to define, and their proofs, especially the volume integrands, somewhat reflect inherent relationship between these two similar algorithms.

Strengths: This paper provides new insights for two common game dynamic systems. The main technique, volume analysis, mainly base on the volume integrand results in [9], is novel and well-explained. The transformation from dynamics of OMWU to a backward finite-difference update for an ODE is quite interesting. As MWU and OMWU are basis for many online learning algorithms, it is good to know their flaws duiring the dynamic processes.

Weaknesses: The main results, the two negative properties, do not undermine the practicalities of the algorithms at all, especially when they are used to find Nash equilibrium. MWU and OMWU are commobly applied mainly due to the convergence of their time average, while this paper is analyzing their day-to-day behaviors.

Correctness: I checked most proofs and they are correct.

Clarity: This paper is well written. The use of volume analysis is easy to follow, and the sketches of proofs are quite intuitive.

Relation to Prior Work: It is clearly discussed how this work differs from previous contributions.

Reproducibility: Yes

Additional Feedback: Some phrases, "volume" in line 163, "diameter" in line 166, "game value" in line 265, are used before definitions, or never defined. \bar{C} in Theorem 3 and Lemma 4 should be defined as \inf_{(x,y)\in U}C(x,y) or \inf_{(x,y)\in V}C(x,y) (actually similar line 197, it may be better to use notation \bar{C}_S). The discussion on RPS in Appendix C.1 is indeed comlicated. I wonder whether it is possible to explore a more general condition for the non-trivial matrix, including both conditions in Theorem 5 and RPS. ========================== Regarding the author's response: My minor concern has been addressed by the author's response.


Review 4

Summary and Contributions: The paper uses volume analysis to understand the behaviour of Multiplicative Weights Update (MWU) and its optimistic variant (OMWU) in zero-sum and coordination games. In particular, the paper: 1) Defines two properties, Extremism (where a system recurrently gets stuck near pure strategies) and Unavoidability (where a set of “bad” points cannot be avoided), and shows that MWU suffers from these properties in the case of zero-sum games. 2) Proves, on the other hand, that OMWU is Lyapunov-chaotic for coordination games, and gives an analysis (distinct from that done in earlier literature) showing that OMWU is stable on zero-sum games.

Strengths: The negative result proved for OMWU in coordination games is novel and would be of interest to the NeurIPS community. The notions of Extremism and Unavoidability that the paper introduces are also sensible and useful. Of secondary importance, the paper was very well-written and could more generally be a useful introduction to the theoretical analysis of zero-sum and coordination games.

Weaknesses: This was a strong paper with no major problems.

Correctness: The motivation and theoretical analysis were logical and sound.

Clarity: The paper was well-written and easy to follow throughout. There were no obvious notational errors or formatting problems.

Relation to Prior Work: Papers published on similar topics were mentioned, and the differences between those and this paper were made clear.

Reproducibility: Yes

Additional Feedback: Just mentioning a few very minor typos: L29 – most -> for most L69 – family -> a family L73 – phenomena -> phenomenon L86 – mild -> a mild L108 – thought -> thought of L111 – condition -> conditions L112 – game -> games L245 – game -> games L251 – boundary -> the boundary L261 – same row -> the same row # post-rebuttal edit After reading the rebuttal I have decided to keep my original rating.

[Author Response · NeurIPS 2020]

First of all, thanks all reviewers for your positive comments on this paper. We will fix the typos and minor errors you
pointed out. We address your questions/suggestions below.

**(1) Empirical Studies.** If the paper is accepted to NeurIPS 2020, we will have one extra page. We will use it to
include more empirical studies beyond the current Figures 1, 3 and 4 (Figures 3, 4 are in appendix). One example is
the following volume-expansion figures in 3D. There are three players, each with two strategies Head and Tail. The
underlying game is a graphical zero-sum game, where each edge-game is a Matching Pennies game: Player $i$ wants to
match with Player $(i + 1)$ but wants to mis-match with Player $(i - 1)$. We start with a small rectangular box around a
Nash equilibrium (leftmost) in the payoff space, using Multiplicative Weights Update (MWU) for each player, and runs
the algorithms for 4500, 9000 and 13500 steps. *The figures are plotted with high resolution; you may magnify them and*
*read them more clearly.*

Indeed, we create more figures and combine them as a video. It demonstrates the evolution more clearly, regarding
volume-expansion, edge-curving and edge-twisting. But we are not allowed to include a link to the video here; it will
be in the final version of this paper. We also run with Optimistic MWU. Since the volume shrinks to a single point
(boringly) as expected, we choose not to include the figures in this short response. They will appear in the final version.

**(2) How may the results, particularly the volume analyses, be translated to the case with "diminishing step-**
**size"? What about other learning dynamics, e.g., those in [1][5][17]?** It is fairly straight-forward to generalize
volume analyses to diminishing step-size; indeed, in all calculations we simply need to change $\epsilon$ to $\epsilon_t$. About the results:

- 18 The unavoidability theorem can be generalized to diminishing step-size when $\epsilon_t = \Omega(1/\sqrt{t})$, since this can
  19 guarantee that the volume expands quickly enough, so that the proof of the theorem can carry through.
- 20 For the extremism theorem, we need unavoidability theorem and a technical lemma (Lemma 10 in Appendix
  21 C). As said, the unavoidability theorem component can carry through with $\epsilon_t = \Omega(1/\sqrt{t})$. For the lemma, we
  22 believe it holds with step-size which is diminishing but not too quickly. But we currently do not see a quick
  23 hack of the current argument to generalize. We believe it can be done with more careful argument.

24 Regarding applicability of volume analyses to other learning dynamics (e.g. those in [1,5,17]): yes it can be applied to
25 those dynamics, as all of them belong to the family of *gradual* update rules which we formulate in this paper. Indeed
26 we are planning to work on some of them in future work.

27 **(3) The main results, the two negative properties, do not undermine the practicalities of the algorithms at all...**
28 **MWU and OMWU are commonly applied mainly due to the convergence of their time average...** We do not have
29 any intent to undermine the practicalities of these algorithms. Rather, our key message is they are not practical in *every*
30 scenario, we need to choose the appropriate ones depending on the applications. On the other hand, we note that there
31 are simple games for which MWU and OMWU both do *not* converge, even in time-averages. One example is the
32 game studied in Kleinberg et al. *Beyond the Nash Equilibrium Barrier* in ICS 2011. They showed instability of simple
33 learning algorithms. We have run simulations to verify that both MWU and OMWU diverge in time-average.

34 **(4) It would be better to compare with the most relevant a prior work [9] with more details.** [9] is the first work
35 in ML venue to *prove* that chaos exists using volume-expansion argument. Our focus in this paper is the new types of
36 negative consequences of volume expansion, namely unavoidability and extremism. We also investigate how volume
37 analyses can be applicable to other learning methods. As we see it, the generalization to OMWU is not straight-forward,
38 and its analysis is interesting from a technical perspective.

39 **(5) The discussion on RPS in Appendix C.1 is indeed complicated. I wonder whether it is possible to explore**
40 **a more general condition for the non-trivial matrix, including both conditions in Theorem 5 and RPS.** We will
41 improve the discussion in Appendix C.1. We also thought about how to generalize to trivial matrices, but we did not
42 see a simple way, unless we introduce further (heavy) mathematical notations. However, this appears to reduce the
43 readability of this paper with relatively little gain, so we opted not to pursue.

[Meta-Review · NeurIPS 2020]

This paper was carefully reviewed and discussed by our reviewers, and I took a look at the work myself. There is an interesting set of tools being developed here, and the application of dynamical systems ideas such as volume analysis for understanding the dynamics of learning in games is very neat. The paper is a little bit out of the usual scope for NeurIPS, but this will be of interest to many members of the community. Please make sure to take into account the detailed comments of the reviewers when preparing your camera-ready.